# Epigenetic Explorations of Neurological Disorders, the Identification Methods, and Therapeutic Avenues

**DOI:** 10.3390/ijms252111658

**Published:** 2024-10-30

**Authors:** Zeba Firdaus, Xiaogang Li

**Affiliations:** 1Department of Internal Medicine, Mayo Clinic, Rochester, MN 55905, USA; firdaus.zeba@mayo.edu; 2Department of Biochemistry and Molecular Biology, Mayo Clinic, Rochester, MN 55905, USA

**Keywords:** neurodegeneration, neuroepigenetics, DNA methylation, histone acetylation, Alzheimer’s disease, amyotrophic lateral sclerosis, Parkinsons’s disease

## Abstract

Neurodegenerative disorders are major health concerns globally, especially in aging societies. The exploration of brain epigenomes, which consist of multiple forms of DNA methylation and covalent histone modifications, offers new and unanticipated perspective into the mechanisms of aging and neurodegenerative diseases. Initially, chromatin defects in the brain were thought to be static abnormalities from early development associated with rare genetic syndromes. However, it is now evident that mutations and the dysregulation of the epigenetic machinery extend across a broader spectrum, encompassing adult-onset neurodegenerative diseases. Hence, it is crucial to develop methodologies that can enhance epigenetic research. Several approaches have been created to investigate alterations in epigenetics on a spectrum of scales—ranging from low to high—with a particular focus on detecting DNA methylation and histone modifications. This article explores the burgeoning realm of neuroepigenetics, emphasizing its role in enhancing our mechanistic comprehension of neurodegenerative disorders and elucidating the predominant techniques employed for detecting modifications in the epigenome. Additionally, we ponder the potential influence of these advancements on shaping future therapeutic approaches.

## 1. Introduction

Neurodegenerative disorders (NDD) present significant challenges and rank among the most critical global health concerns, particularly with the growing aging population worldwide [1]. Neurodegeneration corresponds to any pathological condition primarily affecting neurons. In clinical practice, NDD represent a large group of neurological disorders, with various clinical and pathological characteristics affecting specific subsets of neurons in specific regions of the central nervous system (CNS). Classical NDD include Alzheimer’s disease (AD), Parkinson’s disease (PD), Amyotrophic lateral sclerosis (ALS), frontotemporal dementia (FTD), Huntington’s disease (HD), and Multiple sclerosis (MS) [2]. Although these diseases have different characteristic features, such as different protein aggregates and genetic variations, they all share the common progression mechanism like chronic neuroinflammation, impaired autophagy, loss of proteostasis, mitophagy, telomere dysfunction, and epigenetic alterations [3]. Another common factor that NDD possess is the mutual involvement of protein misfolding with each disease, being associated with a specific protein, a loss of neurons, and synaptic loss [4].

### 1.1. Epidemiology

The impact of NDD is steadily escalating with the aging population, resulting in immeasurable economic and human consequences. Currently, more than 55 million people have dementia worldwide, over 60% of whom live in low- and middle-income countries [5,6]. Furthermore, by 2050, they are projected to become the world’s second leading cause of death, surpassing cancer. While these projections are estimations, when considered alongside the current situation, they undoubtedly underscore the growing public concern surrounding NDD.

The most common and prevalent form of NDD are AD, PD, ALS, HD, and MS [1,7]. The incidence rate of AD varies according to age, with 5% in individuals of ages ranging from 65 to 74 years, 13.1% for 75 to 84 years, and 33.3% of people aged 85 years and older [8]. People younger than 65 can also develop Alzheimer’s dementia. Although prevalence studies of younger-onset dementia in the United States are limited, they are increased in developing countries [6]. The strongest known risk factor for developing PD is age, with 1–3% of the population over 60 affected by PD, rising to 5% over the age of 85 [9]. The highest prevalence rate was seen in Europe and North America and a slightly lower rate in Asian individuals [10]. Recent studies show that the prevalence of ALS is slightly increased and varies between 4.1 and 8.4/100,000 [11]. This increase may be due to demographic changes, as well as to an increase in diagnostic opportunities and prolonging life expectancy due to an improved quality of care. Research based on the American National ALS records reported the ALS prevalence as 5.0/100,000 in 2014 and 5.2/100,000 in 2015 [12]. The analysis of studies from 2010 to 2022 found a pooled incidence rate of 0.48 HD cases per 100,000 person–years, with higher rates in Europe and North America than in Asia [13]. The global prevalence of MS rose to 2.8 million in 2020, a 30% increase from 2013, with a prevalence of 35.9 per 100,000 and an incidence rate of 2.1 per 100,000 annually. MS prevalence has risen worldwide, with only 14% of countries reporting stable or declining rates. Pediatric MS cases have surged to over 30,000, and globally, females are twice as likely to be affected as males, with some regions seeing a 4:1 female-to-male ratio [14].

### 1.2. Pathogenesis

AD is characterized by the deposition of intracellular amyloid β (Aβ) plaques and extracellular neurofibrillary tangles (NFT) of tau in the brain of affected individuals [15]. Several functional interactions have been revealed between Aβ and tau in neural circuit damage and cognitive decline in AD [16]. The key proteins involved in AD pathogenesis are the amyloid precursor protein (APP) and microtubule-associated protein tau (MAPT), encoded by the *APP* gene present on chromosome 21 [17] and the *MAPT* gene present on chromosome 17 [18], respectively. Aβ plaques form through the stepwise cleavage of APP by β-secretase and γ-secretase. Typically, α-secretase or β-secretase initiate the cleavage of the extracellular domain of APP, leading to the production of soluble N-terminal fragments, namely APPsα or APPsβ, and membrane-bound C-terminal fragments, CTFα and CTFβ, respectively [19]. CTFα or CTFβ will further be cleaved by the γ-secretase within the transmembrane domain, releasing the p3 peptide or β-amyloid (Aβ) peptide [20]. After γ-secretase cleavage, both CTFs will additionally release the intracellular domain of APP (AICD) into the cytoplasm. The pathway, characterized by α-secretase-mediated APP cleavage, is termed the non-amyloidogenic pathway, as it inhibits Aβ production. Conversely, the pathway involving β-secretase-mediated APP cleavage, which yields Aβ peptides, is known as the amyloidogenic pathway (Figure 1). The aggregation of Aβ thus forms oligomers and plaques that are toxic to the neurons [19].

Tau, on the other hand, is derived from alternative splicing from the *MAPT* to form soluble protein isoforms [21]. The balance between kinase and phosphatase activity regulates the phosphorylation level of tau and its affinity for microtubule binding [22]. When tau is fully dephosphorylated, it binds tightly to microtubules. However, the activation of tau phosphorylation-associated kinases, such as CDK-5 and GSK-3β, induces the hyperphosphorylation of tau. This hyperphosphorylation then leads to the dissociation of tau protein from microtubules [22]. In physiological conditions, tau undergoes continuous cycles of phosphorylation and dephosphorylation to maintain its functionality. However, if the equilibrium tilts towards abnormal phosphorylation, leading to the hyperphosphorylation of tau, it diminishes microtubule binding and possibly augments tau dissociation. Consequently, this elevates cytosolic tau levels, fostering tau–tau interactions and aggregation (Figure 2) [23]. Aggregated tau can accumulate within neurites and neuronal cell bodies, initially forming insoluble filaments that eventually develop into NFTs [24].

The effect of tauopathy has also been studied in relation to autophagy-mediated neurodegeneration [25]. The tau protein is a microtubule-associated protein primarily found in the axons of neurons, where it is essential for stabilizing microtubule assembly and facilitating axonal transport [26]. In neurons, autophagosomes are generated at the axon tip to carry cellular cargo destined for degradation. These autophagosomes are retrogradely transported along microtubules to the soma, where they fuse with lysosomes to form autophagolysosomes [27]. However, pathological alterations in tau, particularly hyperphosphorylation, diminish its binding affinity for microtubules, resulting in microtubule depolymerization and disrupted axonal transport [28]. Such disturbances can impair autophagy and hinder the clearance of cellular inclusions. Moreover, the accumulation of tau can inhibit IST1 expression, which disrupts the formation of the ESCRT-III complex and impairs lysosomal degradation [29]. This complex interaction between abnormal tau and autophagy creates a detrimental cycle that exacerbates neurodegeneration.

Mitochondrial dysfunction, marked by elevated Ca^2+^ and ROS levels, contributes to the accumulation of phosphorylated tau aggregates [30]. A SOD2 deficiency in the mitochondria has been shown to cause phosphorylated tau accumulation in mice, which can be reversed with antioxidant treatment [31]. Additionally, reducing amyloid aggregation in cells can be achieved by targeting mitophagy both pharmacologically and genetically [32].

Pathologically, PD can be identified by the presence of Lewy bodies which contain α-synuclein protein [33]. The presence of fibrillary α-synuclein suggests the implication of α-synuclein aggregation in the progression of PD [33]. α-synuclein is encoded by a *SNCA* gene present on the long arm of chromosome 4 [34]. Under physiological conditions, natively unfolded α-synuclein in solution can transition into an α-helical conformation within its N-terminal domain when encountering membranes containing acidic phospholipid headgroups and/or exhibiting high curvature. This interaction with membranes typically serves to diminish the likelihood of misfolding into a β-sheet assembly and/or encourages the formation of physiological multimers [35]. Under pathological conditions (Figure 3), focal alterations in pH or Ca^2+^ concentration, alongside other potential factors, are likely to induce aggregation-prone conformations of α-synuclein around membranous compartments [36]. Predominantly, cell death is caused by disruption of nuclear membrane integrity and the release of α-synuclein aggregation, promoting nuclear factors like histones [37]. Once aggregation initiates, α-synuclein may spread to neighboring cells via direct or indirect routes. In individuals with PD, approximately 50–70% of neurons in this area are lost by the time of death, which differs significantly from unaffected individuals [37].

Mutations in genes like *PARK7, α-synuclein, parkin, PINK1*, and *LRRK2* lead to mitochondrial dysfunction, contributing to PD [7]. α-synuclein mutations cause protein aggregation, disrupting mitophagy, while *PINK1* deletion increases oxidative stress in mitochondria [38]. Environmental toxins like MPTP and rotenone also cause mitochondrial dysfunction, mimicking PD in models. The MitoPark mouse, lacking the TFAM (mitochondrial transcription factor A) gene, replicates key PD features, such as dopaminergic neuron loss and motor deficits, underscoring the role of mitochondrial dysfunction in the disease [39].

TDP-43 cytoplasmic inclusions are an almost universal feature of ALS, present in about 97% of cases [40]. TDp-43 encoded by the *TARDBP* gene is present on chromosome 1 [41]. TDP-43 is a widely distributed RNA/DNA-binding protein, predominantly found in the nucleus, where it performs diverse cellular functions such as mRNA splicing, stability, maturation, and transport, along with transcriptional repression. In individuals diagnosed with ALS, TDP-43 undergoes mislocalization to the cytoplasm and experiences substantial post-translational modifications or truncations, or both (Figure 4) [40]. Phosphorylation, ubiquitination, and abnormal cleavage are the primary post-translational modifications identified in TDP-43 protein inclusions, which are believed to be associated with the pathological changes seen in ALS [42]. While the precise pathological significance of phosphorylation remains uncertain, considerable evidence indicates that the hyperphosphorylation of TDP-43 may contribute to neurotoxicity, enhance TDP-43 oligomerization, mislocalization, and seeding [43]. Like phosphorylation, ubiquitination impacts the overall concentration of the TDP-43 protein by activating the UPS and autophagy degradation pathways [44]. A significant hallmark of TDP-43-related neuropathology is the presence of TDP-43 C-terminal fragments (CTFs) produced by caspases and calpain proteases [45]. These PTMs change the characteristics of TDP-43, leading to self-aggregation and the formation of insoluble cytosolic inclusions [46]. Mitophagy, essential for clearing damaged mitochondria, is impaired in ALS, as seen in *SOD1^G39A^* mice, where the neuromuscular junction (NMJ) shows fewer phagosomes compared to wild-type mice. This is linked to the downregulation of mitophagy-related proteins PINK1 and Parkin. In *PINK1-Parkin* double-knockout mice, mitophagy defects result in exacerbated NMJ degeneration, axon swelling, and worsened ALS symptoms [47]. These mice also have elevated levels of the ATP synthase beta subunit, suggesting that defective mitophagy leads to the accumulation of dysfunctional mitochondria at the NMJ [47].

HD is an autosomal, progressive, and dominantly inherited neurodegenerative disorder characterized by neuron degeneration, particularly in the cerebral cortex and striatum, leading to choreatic movement impairments, behavioral changes, and psychiatric symptoms [48]. Although HD’s genetic basis is well-defined, involving CAG repeat expansions in the huntingtin (htt) gene, its complex biological mechanisms include oxidative stress and metabolic and mitochondrial dysfunction. Neurodevelopmental effects include striatal degeneration and neuroinflammatory markers, such as IL-6, VEGF, and TGF-1 [49]. However, HD, as a genetically driven disease, is not the focus of our review, which centers instead on sporadic neurodegenerative diseases.

Multiple sclerosis (MS) is a chronic, inflammatory, and neurodegenerative disease of the central nervous system marked by immune-mediated demyelination and neuronal damage affecting both white and gray matter [50]. This multifactorial disorder results from complex genetic and environmental interactions. The primary pathology of MS involves the accumulation of demyelinating lesions or plaques, driven by T-lymphocytes and macrophages, leading to the progressive damage of the myelin sheath [51]. This disruption of nerve signal transmission causes a wide range of symptoms, including sensorimotor dysfunction, visual disturbances, fatigue, cognitive impairments, and emotional challenges, reflecting the extensive impact on the CNS [51]. The pathogenesis of MS involves intricate immune interactions, starting with antigen-presenting cells that activate T lymphocytes, which then differentiate into proinflammatory subsets such as Th1, Th2, and Th17. These subsets release various cytokines-Th1 cells which promote inflammation through IFN-γ and TNF-α, Th2 cells which have anti-inflammatory roles via IL-4 and IL-13, and Th17 cells which contribute to neurodegeneration through IL-17 and related cytokines [51]. Due to the immune-centered complexity of MS, this review will instead focus on neurodegenerative diseases driven by protein aggregation, excluding MS from further discussion.

Although only a minor subset (10%) of these disorders can be linked to clearly defined genetic factors [52], the majority of cases are sporadic and arise from a complex interplay between genetic and environmental elements [53,54]. These external influences, such as diet, lifestyle, exposure to toxins, and stress, can modify gene expression through epigenetic mechanisms like DNA methylation, histone modification, and non-coding RNA activity. For example, chronic exposure to pesticides has been linked to an increased risk of Parkinson’s disease by promoting oxidative stress and altering DNA methylation patterns in genes associated with dopaminergic neuron survival [9]. Similarly, dietary factors like a high-fat intake have been shown to dysregulate histone acetylation, impacting genes involved in mitochondrial function and increasing vulnerability to neurodegeneration in AD [55]. Physical exercise, conversely, exerts a protective effect by enhancing brain-derived neurotrophic factor (BDNF) expression through epigenetic changes, promoting synaptic plasticity and neuronal survival [56].

Nonetheless, because of their sporadic occurrence, the role of epigenetic alterations in the emergence of neurodegenerative conditions has gained enough interest over the past decade. The epigenetic makeup provides a framework in which environmental factors can engage with an individual’s genetic constitution. Additionally, the progress in high-throughput technologies for genomic, transcriptomic, and epigenomic analysis has expanded the scope of understanding epigenetic modifications in the context of neurodegeneration, offering the potential for a comprehensive view of their interplay with various ‘omic’ layers. Concurrent research has significantly advanced our knowledge of the cellular mechanism, thereby paving the way for the exploration of novel therapeutic targets.

This review primarily delves into the understanding of neuroepigenetics and deciphering the complex interplay of epigenetic changes in neurological disorders like ALS, AD, and PD. Our examination encompasses the methodologies employed to identify these modifications. Drawing from ongoing discussions, we aim to suggest potential directions for future research and assess the therapeutic possibilities within this domain.

## 2. Neuroepigenetics

The term “epigenetics” was first used by Conrad Waddington in the 1940s to refer to his logical conclusion that, during embryonic development, a number of mechanisms must exist above (or “epi”) the level of genes in order for identical genes to be expressed differently in various cell types and contexts to determine the cell fate [57]. In its most classical definition, “epigenetics” refers to all heritable changes in gene expression that are not coded in the DNA sequence and solely affect the phenotype without modifying the genotype [58]. These changes are primarily caused by DNA methylation or hydroxymethylation, histone post-translational modifications, and changes in nucleosome positioning; these are collectively termed as ‘chromatin remodeling’ [59,60]. Histone variations, microRNAs (miRNAs), and long non-coding RNAs (lncRNAs) have all been found as additional epigenetic factors in recent investigations [61,62].

As stated above, epigenetic changes are traditionally used to pass on acquired non-DNA-sequence information to progeny cells. At first appearance, these themes appeared to have minimal significance to the human postnatal or adult brain, which contains a significant number of postmitotic and highly differentiated cells and in which the majority of neurons grow, differentiate, and exit the cell cycle permanently many weeks before birth. However, a reappraisal of the possible significance of epigenetics in brain development and disease has been driven by three recent significant breakthroughs. First, human investigations have shown that the epigenetic landscape remains ‘plastic’ throughout all stages of brain development and aging, and that continuing dynamic regulation exists even in neurons and other postmitotic brain elements [63,64]. Second, aberrant chromatin organization and function have been associated not only with cases of neurodevelopmental defects in early childhood, but also to a subgroup of adult-onset hereditary neurological conditions [65,66]. Third, a fast-developing group of chromatin-modifying medicines have demonstrated surprising therapeutic results for a wide spectrum of degenerative and functional nervous system [67,68]. These three lines of research have now converged, sparking considerable interest in the chromatin-associated mechanisms of neurological disease and laying the groundwork for a new discipline known as ‘neuroepigenetics’ [69,70].

Contrary to how epigenetics generally works to pass on acquired non-DNA sequence information, neuroepigenetics, which deals with neurons that do not divide, reveals that these mutations are not passed to daughter cells. Despite this key distinction between traditional epigenetics and neuroepigenetics, the same transcription factors and epigenetic processes are involved to regulate gene expression in both contexts. Epigenetic mechanisms in neurons are used for synaptic plasticity, the acquisition and consolidation of memory, and circuit regulation [71]; as a result, the failure of these systems might impair fundamental network-related cognitive function and contribute to NDD [72]. Among the most significant regulators are chromatin remodeling enzymes, which alter the chromatin structure to facilitate or repress transcription in response to various cellular stimuli. In the context of NDD, these enzymes, including the SWI/SNF, NuRD, CHD, ISWI, and Polycomb complexes, play crucial roles in controlling the expression of genes involved in neuronal function, survival, and synaptic plasticity [73].

## 3. Epigenetic Modifications

Epigenetic mechanisms regulate gene expression by modifying the structure of DNA, making certain areas more or less accessible to transcriptional machinery. DNA is packed into the chromosome using chromatin, which is made up of 147 DNA base pairs securely wrapped around the histone proteins, including two copies of each core histone, H2A, H2B, H3, and H4, which are organized to create the nucleosome [74]. Furthermore, the nucleosome structures are stabilized by the linker histone H1, which binds to the DNA between the nucleosomal core particles, and numerous nucleosomes constitute the chromatin material in a cell [74]. Epigenetic mechanisms control changes in chromatin accessibility, which can activate or repress gene transcription. Particularly, the heterochromatin structure forms an inactive state of the chromatin and represses the transcription of DNA, whereas the euchromatin forms an active state and allows gene expression [74]. The accessibility of chromatin is modulated by internal changes in the DNA itself or by the post-translational alteration of the histone proteins within the nucleosome [74]. Common epigenetic changes including DNA methylation and histone modification are discussed here briefly.

### 3.1. DNA Methylation

DNA methylation, which generally suppresses gene expression (Figure 5), is a crucial modulator of memory formation and storage in the brain [75]. DNMT1, DNMT3A, and DNMT3B have been linked to the experience-dependent alteration of brain function in adults [76]. Moreover, in rodents, DNMTs are associated with contextual fear conditioning and hippocampal-dependent learning and memory [77].

Although methylation was initially considered as a persistent marker of gene silencing, later we understand that changes in the DNA methylation status of synaptic plasticity and memory-associated genes can be both quick and reversible [78]. Ten–eleven translocation protein 1 (TET1), TET2, and TET3 are methyl cytosine dioxygenases that catalyze the conversion of 5-methylcytosine to 5-hydroxymethylcytosine [79]. TET1 catalyzes the DNA demethylation and promotes the activation of genes necessary for memory formation and consolidation, including activity-regulated cytoskeletal-associated protein (ARC), brain-derived neurotrophic factor (BDNF), early growth response protein 1 (EGR1), fibroblast growth factor (FGF), the transcription factor FOS, the scaffolding protein HOMER1, and nuclear receptor subfamily 4 group A member 2 (NR4A2) in mice [80]. Therefore, both DNA methylation and demethylation, as well as the rapid cycling between the two states, may be dysregulated in the decreased cognition that is linked with neurodegeneration [81].

BDNF plays a crucial role in synaptic plasticity and is essential for memory and learning, particularly influencing neuroplasticity in the hippocampus and prefrontal cortex [82]. Recent research suggests that dysregulated BDNF contributes to the development of several major disorders, including AD and depression, with epigenetic processes playing a key role [83]. Additionally, studies have linked BDNF methylation at CpG regions to cognitive impairments and psychiatric conditions [84].

Egr1 (Zif268), a transcription factor and immediate–early gene from the Egr family, is affected by aging [85]. It is rapidly activated during long-term potentiation (LTP) and in key brain regions involved in learning, following specific learning experiences. Egr1 is essential for maintaining the long-term stability of spatial representations in the hippocampus, a process that becomes impaired in aging rats [85]. The mechanisms driving age-related changes in gene transcription remain unclear but may involve altered epigenetic modifications like DNA methylation and histone changes. For instance, the abnormal methylation of the Arc gene has been shown to impair memory function in aged rats [86], while changes in histone methylation around the BDNF and Egr1 genes have been observed in the aged hippocampus [87]. Egr1 transcription and DNA methylation in the CA1 and dentate gyrus regions of the hippocampus have been studied in both normal adult and memory-impaired aged animals, revealing specific age-related changes in Egr1 gene activity and DNA methylation patterns in certain hippocampal subregions [88].

DNA methylation is an epigenetic process that produces direct modifications of DNA by adding a methyl group to cytosine. DNA methyltransferases (DNMTs) catalyze DNA methylation by shifting a methyl group from S-adenosyl methionine (SAM) to the fifth carbon of a cytosine residue, resulting in 5methyl Cytosine (5mC) [74]. The de novo DNMTs generate new methylation marks, whereas the maintenance DNMT (DNMT1) preserves already marked methylation on DNA by methylating the opposite DNA strand [74]. During DNA replication, DNMT-1 removes the methyl group from the parent strands of DNA and transfers it to the newly synthesized daughter strand [89]. The presence of significant amounts of DNA methyltransferases in adult mammalian brain (having postmitotic cells) suggests an additional role for DNMT in the brain [89], which is beyond the scope of this article.

Mitoepigenetics has historically been overlooked due to methodological challenges, and its existence is still debated. Mitochondrial DNA (mtDNA) is distinct from nuclear DNA; it lacks CpG islands and is organized into nucleoprotein complexes called nucleoids without histones. A significant 2011 study demonstrated the presence of mtDNA epigenetic marks by identifying both methylated (5-mC) and hydroxymethylated (5-hmC) cytosines in mitochondrial DNA from human and mouse cell lines, along with the enzyme mitochondrial DNA methyltransferase 1 (DNMT1), which is responsible for adding methyl groups to cytosines [90].

Although mtDNA methylation and hydroxymethylation levels are much lower compared to nuclear DNA, they may still play a critical role in regulating mtDNA replication and transcription. Numerous studies have associated these methylation patterns with a variety of human diseases including neurodegenerative diseases.

Further studies revealed abnormal mitochondrial DNA (mtDNA) methylation patterns and elevated DNMT3A levels in ALS mouse models carrying *SOD1* mutations, including the increased methylation of the 16S rRNA gene and reduced methylation in the D-loop region [91]. Recent research in ALS patients and presymptomatic *SOD1* mutation carriers also identified decreased D-loop methylation, which was strongly correlated with a higher mtDNA copy number [92]. This hypomethylation may serve as a compensatory mechanism to enhance mtDNA replication in response to oxidative stress [92]. These findings collectively suggest that disrupted mtDNA methylation is an early molecular event in ALS, particularly in individuals with *SOD1* mutations. Studies on Alzheimer’s disease (AD) revealed a slight, non-significant increase in 5-hmC levels in the mtDNA of brain samples from late-onset AD patients compared to controls [93]. Another study found increased mtDNA D-loop methylation in the entorhinal cortex of AD patients, with higher levels in early disease stages than in later ones. This pattern was confirmed in transgenic AD mice (APP/PS1) as the disease progressed [94]. In a separate analysis of blood samples from 133 late-onset AD patients and 130 controls, a significant reduction in D-loop methylation was observed in AD patients [95]. Decreased D-loop methylation was observed in the substantia nigra of PD patients compared to healthy controls, suggesting that D-loop methylation might vary across different stages of neurodegeneration [94]. Limited research has examined methylation in other mtDNA regions in neurodegenerative diseases. Notably, reduced MT-ND1 methylation was found in the entorhinal cortex of Alzheimer’s disease (AD) patients, while no differences in MT-ND6 methylation were observed between PD patients and controls [94].

### 3.2. Histone Methylation

Histone methylation is responsible for both transcriptional activation and repression, depending on the changed amino acid residue. This modification occurs mainly on arginine and lysine residues. Additionally, these residues could be methylated multiple times, giving different signals depending on how many times the residue is methylated, making its analysis difficult. In this regard, the literature has shown that lysine residues can be methylated even three times; meanwhile, arginine residues can only be methylated twice [96]. Generally, monomethylation is linked with transcription activation, whereas trimethylation has been connected to transcription repression (Figure 5) [97]. The dimethylation of H3K36 has been related to RNA polymerase II elongation during transcription [72]. Also, the dimethylation of H3K79 is particular of promoter regions stimulating a permissive chromatin for local transcription. Conversely, the modifications associated with transcriptional repression are performed on H3K9 and H3K27 residues [72]. Histone methylation plays a crucial role in gene regulation, with specific residues associated with transcriptional outcomes. The methylation of histone H3 at lysine 4 (H3K4), lysine 14 (H3K14), lysine 36 (H3K36), lysine 79 (H3K79), and arginine 17 (H3R17) is linked to gene activation, while methylation at lysine 9 (H3K9), lysine 27 (H3K27), and histone H4 at lysine 20 (H4K20) is associated with gene repression [98].

### 3.3. Histone Acetylation

Each histone protein has a core globular domain and an N-terminal tail with several possible modification sites. In this context, the amino acid residues of histones have been characterized with a number of post-translational modifications include acetylation, methylation, phosphorylation, sumoylation, and ubiquitination [99]. The main amino acid residues where these modifications occur are arginine, lysine, serine, and threonine [100]. Depending on the modification site, they are associated with the activation or repression of gene transcription which strongly suggests the existence of a histone code [101,102]. Furthermore, these modifications are dynamic as they are actively added and removed by histone-modifying enzymes in a site-specific manner, which is crucial for coordinated transcriptional control. Here, we will briefly explore various types of histone modifications.

Histone acetylation is the widely studied histone modification and is strongly connected with memory formation. Histone acetyltransferases (HATs) catalyze the addition of acetyl groups to histones. Histone acetylation relaxes histone–DNA connections, resulting in a more open configuration that allows transcriptional machinery to reach gene promoters [103] and upregulate transcription (Figure 5). Chemically, when an acetyl group (COCH_3_) is added to an amino terminal residue, the positive charge of histones is reduced, resulting in a minor repulsion with DNA and lower chromatin compaction [104].

The first study to link histone modifications to cognition discovered that novel taste learning in mice activates ERK-MAPK signaling, which in turn activates HATs and drives histone acetylation in gene promoters in the insular cortex, a brain region critical for the formation of novel taste memories [105]. This study suggested histone acetylation to be involved in single-trial learning and the formation of long-term memories. In a contextual fear-conditioning paradigm, HDAC inhibitors that maintain histone tail acetylation boost mouse performance [106]. Mice expressing the dominant-negative form of the CREB-binding protein (CBP; a HAT and crucial binding partner of CREB) and mice deficient in CBP showed impairment in long-term memory, which was treated by HDAC inhibitors [107].

### 3.4. MicroRNAs (miRNAs)

miRNA are small, non-coding RNA molecules that regulate gene expression at the post-transcriptional level by binding to the 3′ untranslated regions (UTRs) of target mRNAs, leading to either their degradation or inhibition of translation. In the context of NDD, numerous miRNAs become dysregulated, affecting critical cellular processes such as the synaptic function, neuroinflammation, and apoptosis [108]. For instance, miR-29a/b and miR-34a have been linked to AD, as they play roles in amyloid-beta (Aβ) accumulation and tau hyperphosphorylation, both of which are key pathological features of the disease [109]. miR-29a/b specifically targets the BACE1 gene, which encodes the enzyme responsible for Aβ production [110]. When these miRNAs are downregulated, BACE1 expression increases, leading to elevated Aβ levels and subsequent plaque formation [109,110]. In PD, miR-133b is essential for the regulation of dopaminergic neurons [111]. The reduced expression of miR-133b in the substantia nigra impairs the maintenance of these neurons, contributing to the neurodegeneration observed in PD patients. This dysregulation impacts both neuronal survival and function, accelerating disease progression [111].

### 3.5. Long Non-Coding RNAs (lncRNAs)

lncRNAs are longer transcripts (>200 nucleotides) that regulate gene expression at multiple levels, including chromatin remodeling, transcriptional control, and post-transcriptional processing [112]. The lncRNA BACE1-AS (BACE1-antisense) regulates the *BACE1* gene, increasing its expression, which in turn promotes Aβ production and accumulation [113]. lncRNA NEAT1 has been linked to ALS, where its overexpression promotes the formation of paraspeckles (nuclear bodies), leading to impaired RNA processing and contributing to motor neuron degeneration [114].

### 3.6. Transcription Factor

Several transcription factors, including REST, SWI/SNF, and Polycomb proteins, are key regulators of epigenetic remodeling in the brain. These gene-silencing factors influence histone acetylation and methylation, controlling the expression of genes associated with synaptic function, such as those encoding synaptic vesicle proteins, receptors, and ion channels. REST and Polycomb proteins play essential roles in brain development and function. Their dysregulation has been linked to neurological diseases, highlighting their importance in both normal brain processes and disease states. Restrictive element 1 silencing transcription factor (REST) is widely expressed during embryogenesis, where it represses neuron-specific genes in pluripotent stem cells and neural progenitors. These genes are essential for synaptic plasticity and structural remodeling during development [115]. REST regulates the expression of genes encoding synaptic vesicle proteins, ion channels, neurotransmitter receptors, and microRNAs (miRNAs) that influence networks of non-neuronal genes. By controlling these key elements, REST plays a crucial role in neural development, ensuring proper timing for neuronal differentiation and function. In a healthy aging brain, REST functions to suppress genes associated with apoptosis and oxidative stress. However, the depletion of REST has been directly linked to the onset and progression of Alzheimer’s disease. REST represses gene transcription by binding to restrictive element 1 (RE1) in target gene promoters and recruiting co-repressors like CoREST and SIN3A. These complexes attract histone deacetylases (HDACs), leading to chromatin tightening and gene silencing. REST also recruits G9A, a histone methyltransferase, and other proteins like LSD1 and MECP2, which further enhance gene repression. Even after REST depletion, co-repressors maintain repression [116]. Recent evidence indicates that REST can both repress and activate target genes. REST has been found to bind the short form of TET3 in neurons, directing it to promoters of REST target genes to generate 5-hydroxymethylcytosine, leading to gene activation. Additionally, REST interacts with NSD3 and other H3K36 methyltransferases, adding H3K36me3 marks associated with gene activation [117]. This suggests that REST’s role in regulating neuronal gene expression is more complex than previously thought. Since REST interacts with DNA methylation machinery, it is linked it to both histone modifications and DNA methylation, impacting epigenetic regulation and drug discovery efforts.

## 4. Modern Techniques to Identify Epigenetic Changes

The fast-paced advancements in epigenetics demand advanced technologies. In addition to established methods like bisulfite sequencing for DNA methylation analysis and ChIP assays for detecting chromatin modifications, a range of novel tools has emerged, leveraging traditional platforms to facilitate significant scientific breakthroughs [118,119]. The array of epigenetic technologies encompasses novel methodologies that enable access at the single-cell level, facilitating high-throughput genomic and epigenomic profiling with ever-increasing precision. Within this section, we will discuss both conventional methods and cutting-edge technologies for deciphering epigenetic codes, offering a thorough overview of the methodologies employed in the field of epigenetics.

### 4.1. Epigenetics Technologies in DNA Methylation

Over the past few decades, the field of DNA methylation has seen significant advancements, leading to the widespread adoption of established techniques for assessing the DNA methylation status across various levels and dimensions (Figure 6). Therefore, the critical task of selecting an appropriate method becomes essential when aiming to address specific scientific questions. DNA bisulfite treatment is a key method for detecting DNA methylation, especially 5mC [120]. This method entails subjecting genomic DNA to sodium bisulfite, prompting the deamination of unmethylated cytosines into uracil while preserving methylated cytosines. This preservation enables the differentiation of 5mCs from unmethylated cytosines across genomic DNA [120]. Widely acknowledged as the benchmark for DNA methylation detection, bisulfite genomic sequencing underpins numerous methodologies designed for the analysis of bisulfite-treated DNA.

Various methods are accessible for locus-specific analysis, including the direct sequencing of bisulfite PCR products, sub-cloning sequencing, methylation-specific PCR (MSP), and pyrosequencing analyses [121]. The MSP assay utilizes bisulfite-converted DNA as the template for PCR processes, enabling the assessment of the methylation status in specific loci. Specific primers are employed to recognize methylated or unmethylated DNA templates. Pyrosequencing, a sequencing-oriented technique, enables the continuous tracking of nucleotide incorporation by detecting pyrophosphate through bioluminometric methods in real-time [122]. Utilizing pyrosequencing for DNA methylation analysis, which combines bisulfite conversion with the pyrosequencing protocol, provides a dependable and accurate quantification approach for assessing the methylation status of a particular gene of interest. This occurs with a high level of quantitative resolution after PCR amplification [122]. Understanding the widespread distribution of DNA methylation profiles underscores the significance of these complex patterns and their impact on biological functions.

In addition to bisulfite conversion, the global DNA methylation status can be easily assessed using hybridization-based microarrays like Illumina HumanMethylation450 (HM450K) or through high-throughput Next-Generation Sequencing (NGS) methods, such as whole-genome bisulfite sequencing (WGBS) [123]. The Illumina HumanMethylation450 (HM450K) utilizes predesigned probes for both methylated and unmethylated CpGs, allowing it to interrogate over 450,000 methylation sites, effectively covering the majority of CpG islands. This platform has found widespread use in human methylomic studies [123]. For a thorough examination of the global DNA methylation status, encompassing all CpG information, Whole-Genome Bisulfite Sequencing (WGBS) stands as the standard profiling method. It allows for the comprehensive assessment of methylation states, including low CpG-density regions such as intergenic regions, partially methylated domains, and distal regulatory elements [124]. WGBS provides a comprehensive and unbiased assessment of the DNA methylation status, but the extensive data it produces necessitate an advanced bioinformatics analysis. Additionally, the budgetary challenges become significant, particularly when dealing with multiple replicative samples in WGBS testing [125].

The platform of single-cell genomics is leveraged in single-cell bisulfite sequencing, providing unparalleled insights into individual cells [126]. Several single-cell bisulfite sequencing (scBS-seq) methods are used to assess the single-cell DNA methylome, including single-cell reduced-representation bisulfite sequencing (scRRBS), single-cell whole-genome bisulfite sequencing (scWGBS), single-nucleus methylcytosine sequencing (snmC-seq), and single-cell combinatorial indexing for methylation (sci-MET) [127]. The first single-cell multiomics analysis combining the DNA methylome and transcriptome was achieved through scM&T-seq, which isolates and amplifies gDNA and RNA from the same cell using G&T-seq and scBS-seq [127]. scMT-seq developed by Hu et al. uses micropipetting to isolate nuclei and applies scRRBS and modified Smart-seq2 for methylome and transcriptome data generation [128]. scTrio-seq profiles the genome, methylome, and transcriptome, using scRRBS for methylation data. However, scMT-seq method has several limitations that could be addressed with future technological advancements. For instance, scRRBS covers only about 1% of CpG sites across the genome, whereas single-cell whole-genome bisulfite sequencing can capture up to 48.4% of CpG sites, allowing for a more comprehensive analysis of DNA methylation and RNA transcription [129]. Another significant limitation is the high rate of allele drop-out, making it less effective for analyzing genes with differential expression between alleles due to varying methylation patterns [128].

Single-cell bisulfite sequencing also has challenges like DNA degradation from bisulfite treatment, low input, and difficulty in assessing methylation differences between individual cells [126]. The drawbacks have been addressed through the integration of post-bisulfite adaptor tagging (PBAT). This involves conducting bisulfite treatment before adapter tagging, allowing for the use of minimal DNA input. Following this, PCR amplification is executed, and then a thorough sequencing is carried out at the single-cell level [130]. Both conventional methylomic profiling techniques such as single-cell WGBS (scWGBS) and single-cell RRBS (scRRBS) are well adopted for single-cell methylation analysis [126]. Enhancements in several areas could improve methylation detection across both alleles like optimizing bisulfite treatment conditions to minimize DNA degradation, refining purification methods to reduce stochastic DNA loss, and increasing the adapter ligation efficiency to capture more DNA fragments.

Taking advantage of third-generation long-reading sequencing technologies, recent progress in epigenetics research involves incorporating Pacific Biosciences’ (PacBio) single-molecule real-time sequencing (SMRT) [131] and Oxford Nanopore Technologies’ (ONT) nanopore sequencing [132]. Both methods can sequence native DNA, eliminating concerns related to DNA degradation caused by bisulfite conversion. Since PCR amplification is not involved, these methods are more suitable for analyzing bulk samples rather than dealing with small amounts of DNA. The absence of PCR amplification helps preserve the integrity of the methylation information and makes these approaches valuable for applications where accurate DNA methylation profiling is crucial.

PacBio SMRT sequencing directly identifies base modifications by observing the polymerase kinetics while incorporating various fluorescently labeled nucleotides into DNA double-strands during synthesis [133]. In contrast to WGBS or HM450K, SMRT sequencing excels in its capability to concurrently reveal both the nucleotide sequence and major DNA methylation patterns, encompassing 5mC, 5hmC, 6mA, and 4mC [134]. While SMRT sequencing identifies base modifications through kinetic changes, its accuracy can be affected by varying sensitivities due to signal-to-noise ratios specific to each modification type. For instance, 6mA and 4mC generate robust kinetics signals, while the signal for 5mC is relatively weak. Consequently, SMRT sequencing is recommended for detecting bacterial genomes, where 6mA and 4mC are common and often concentrated on specific motifs [134].

Nanopore sequencing can discern DNA methylation patterns by threading single-stranded DNA through a biological nanopore. The recorded ion current deviations through the pore correspond to specific base modifications [135].

### 4.2. Epigenetics Technologies in Histone Modifications

Advanced technologies in epigenetics contribute to a deeper understanding of the functional interactions among histone modifications, the transcriptional machinery, and their roles in biology. In recent years, the study of histone modifications in the field of epigenetics has greatly expanded (Figure 7). The widely employed technique for identifying and quantifying chromatin modifications and interaction patterns is the chromatin immunoprecipitation (ChIP) method [136]. The ChIP assay relies on antibodies with the ability to recognize specific histone modification markers or epigenetic modulators, paired with specific DNA fragments.

This approach facilitates the assignment of locus-specific functions to histone modifications or transcription factor complexes, which can have direct or indirect effects on the chromatin structure and the efficiency of the transcriptional machinery [137]. When dealing with a specific histone modification and DNA regulatory region, ChIP, followed by traditional PCR or quantitative real-time PCR (qRT-PCR), can expose the enrichments of targeted histone modifications or the binding affinity of a remodeling complex to the designated DNA area [137]. In contrast, when specific modification patterns are unclear, sequencing-based ChIP methods such as ChIP-chip or ChIP-seq allow for the simultaneous analysis of protein–DNA binding events and histone modification enrichment across a multitude of loci [138].

ChIP-seq is widely used in studies exploring the connection between transcription factors and nucleosome architecture, influencing chromatin dynamics. The analysis of dynamic interactions between methylation and chromatin is a complex task. Researchers devised the ChIP-bisulfite methylation sequencing (ChIP-BMS) approach to effectively evaluate the methylation status of ChIP-pulled DNA using specific antibodies for histone markers or transcription factors [139]. A technique developed by Statham et al., called BisChIP-seq, utilizes the high-throughput sequencing of bisulfite-treated chromatin immunoprecipitated DNA. This approach allows for a comprehensive examination of the genome-wide association between DNA methylation and crucial epigenetic regulators [140]. Despite sharing similar theoretical principles, ChIP-BMS is utilized for detecting methylation at specific loci, whereas BisChIP-seq is designed for a broader, global profiling of methylation. Droplet-based chromatin immunoprecipitation (Drop-ChIP) sequencing enables the measurement of histone modifications, such as H3 di- and tri-methylation, at a single-cell resolution [141]. It uses microfluidic devices to encapsulate individual cells in droplets with lysis detergent and micrococcal nuclease, producing nucleosomes that are barcoded and pooled for ChIP-seq analysis. Despite its innovation, Drop-ChIP provides limited DNA methylome coverage (~800 peaks per cell) [141].

Computational methods for integrating single cell multiomics data are still developing, despite advancements in experimental protocols. Effective methods are needed to address these discrepancies and develop better multiomics integration models. Current approaches are limited to integrating two omics layers, but as new technologies emerge, methods capable of integrating three or more omics layers will be crucial for a deeper understanding of regulatory relationships across different omics data.

### 4.3. Advanced Methods for Characterizing Three-Dimensional Chromatin Organization and the Epigenome

The elucidation of the three-dimensional (3D) organization of the chromatin structure through structural biology is pivotal in understanding epigenetic regulation. A groundbreaking technology, Hi-C, enables the capture of the 3D chromatin structure by producing high-resolution contact maps of the genome on a large scale [142]. Hi-C utilizes proximity ligation and deep sequencing to map genome-wide chromatin interactions, revealing compartmentalized regions of active and inactive chromatin, supporting the concept of nuclear territories [143]. It supports the fractal globule model, which describes chromatin as unknotted, self-similar, and organized in a hierarchical, highly compartmentalized structure within the nucleus [144]. Within these compartments, homotypic interactions dominate, with heterochromatin often positioned at the nuclear periphery. Hi-C also identified topologically associating domains (TADs), where interactions are more frequent within a TAD than between them [145]. This technique has connected changes in the chromatin architecture to genetic disorders like F-syndrome, polydactyly, and brachydactyly [146].

However, Hi-C has several limitations. The use of formaldehyde for crosslinking can hinder the access of restriction enzymes to DNA due to nonspecific protein crosslinking, which lowers the resolution and increases the noise. Additionally, the use of restriction enzymes introduces sequence bias by targeting specific DNA sequences. Moreover, traditional Hi-C only captures pairwise interactions, overlooking the more complex, multiway interactions within the nucleus. To address these issues, newer versions of Hi-C have been developed that bypass the need for crosslinking, restriction enzymes, or ligation, improving the accuracy and resolution. The chemical-crosslinking-assisted proximity capture (CAP-C) method was developed to overcome the biases of protein–DNA crosslinking. CAP-C uses dendrimers and UV irradiation for consistent DNA fragmentation, improving precision and reducing noise [147]. This technique allows for the high-resolution detection of transcription-related changes in the chromatin structure, revealing that transcription initiation mainly influences local chromatin organization, and inhibiting transcription reduces chromatin interactions. Hsieh et al. developed Micro-C, a method that overcomes the sequence bias of restriction enzymes by using micrococcal nuclease (MNase) to cleave DNA in nucleosome linker regions. This approach creates higher-resolution chromosomal folding maps by capturing detailed chromatin conformations without the need for additional enrichment processes, unlike Hi-C [148].

Integrating the 3D structure of chromatin with maps of protein-binding and epigenetic modifications is crucial for fully understanding gene regulation. The initial mapping of protein–DNA associations rely on pulldown and sequencing techniques to isolate DNA-binding proteins and their bound DNA. Chromatin immunoprecipitation sequencing (ChIP-seq) has been key for the genome-wide mapping of chromatin marks and protein-binding sites, initially offering one-dimensional views [149]. Fullwood et al. developed ChIA-PET, which combines sonicated chromatin–protein complexes with ChIP to ligate and sequence DNA fragments based on proximity, elucidating the 3D interactome involving specific proteins [150]. Mumbach et al. introduced HiChIP, an improved method that captures long-range chromatin interactions associated with specific proteins more efficiently and with less input material [151].

While histone modifications and chromatin architecture are typically determined through distinct assays, the integration of ChIP-seq and Hi-C data allows for the exploration of correlations between chromatin organization, such as chromatin interaction compartments and topologically associated domains, with specific histone modifications [152,153]. This predictive model system unveils significant networks linking the chromatin structure to the underlying epigenetic mechanisms that regulate dynamic gene expression. A notable development introduces the Methyl-Hi-C molecular assay, enabling the simultaneous capture of chromosome conformation and the DNA methylome at the single-cell level [154]. The advent of this novel technology enables the simultaneous characterization of cell type-specific chromatin organization and epigenome within intricate tissues. This substantial progress propels investigations into chromosome structural changes and their biomedical functions forward, offering a unique single-cell perspective.

In the next section, we review histone modification including acetylation and methylation linked to neurological disorders, as well as the mechanism of epigenetic dysregulation that affects cognition and causes neuronal death in NDD with a focus on AD, PD, and ALS. Finally, we emphasize the potential efficacy of novel therapeutic strategies that target epigenetic machinery to alleviate the symptoms of NDD.

## 5. Dysregulation of Histone Modifications in Alzheimer’s Disease

AD and related dementia conditions are associated with aberrant histone acetylation [155]. Studies have shown that the histone acetyltransferase activity of CBP (KAT3A) is required for neurogenesis and memory formation and have also suggested that increasing CBP expression could be a viable treatment for AD [107,156,157]. In 3xTg-AD mice, CREB activation and phosphorylation was downregulated and CBP gene transfer restored the CREB activity, which increased the expression of its target gene BDNF and improved learning and memory impairment in mice [158]. Reduced levels of phosphorylated CREB, as well as its cofactors CBP and p300 was also reported in an APPswe/PS1ΔE9 mutant AD mice model [159]. On the contrary, another study reported increased p300 acetyltransferase activity in the hippocampal area of AD patients [160]. Additionally, elevated levels of CBP/p300 and TRAPP were seen in a transcriptomic analysis of the lateral temporal lobe of AD patients, which mediate the acetylation of H3K27 and H3K9 in disease-related genes [161].

These disparities could be explained by methodological variations or risk factors that differ between humans and mice, but further study is needed to fully understand the mechanism. The protein levels of HDAC1 and HDAC3 remain unchanged in a mouse model of AD as well as in AD patients. However, HDAC2 expression was increased, which lowers the histone acetylation of genes involved in learning and memory and suppresses their expression [162]. Further, the knockdown of HDAC2 increased the histone acetylation mark (H4K12ac), leading to the expression of these genes [163]. Additionally, there were significant spikes in the HDAC6 protein level, which interacts with tau to promote tau phosphorylation and accumulation in the AD brain [164]. The downregulation of endogenous HDAC6 levels alleviates Aβ-induced aberrant mitochondrial trafficking and memory deficits in a mouse model for AD [165]. Sirtuins like SIRT1 and SIRT2 also suppress Aβ expression and have emerged as forthcoming pharmaceutical targets for the treatment of AD [166,167].

Apart from acetylation, several histone methylation marks were also found to be elevated in AD patients, suggesting that the expression of histone methyltransferase (KMT2C, KMT2D, SETD1A, and SETD1B) was also altered [155,168,169]. H3K4 HMT was significantly upregulated in AD patients as well as transgenic mouse with human P301S, resulting in higher levels of H3K4me3 in the nuclear fraction of the prefrontal cortex lysates [170]. Histone methyltransferases including EHMT1 and EHMT2 were also substantially increased in the late-stage familial AD mouse model and in AD patients, which in turn elevated the repressive mark H3K9me2 selectively in glutamate receptor genes in the prefrontal cortex and suppressed the transcriptions. Furthermore, the downregulation of EHMT1/2 reduced the H3K9me2 level, resulting in the normal expression of the glutamate receptor and synaptic function in the prefrontal cortex and hippocampus [171].

Studies have found that AD patients exhibit lower levels of H3K4me3 and higher levels of H3K27me3 compared to healthy individuals [172]. These changes in histone methylation have led to several hypotheses regarding their role in AD pathogenesis. One hypothesis posits that the direct methylation of the MAPT gene may facilitate tau aggregation, contributing to neurodegeneration [173]. Furthermore, altered histone methylation has been associated with impaired autophagy, which can worsen AD neuropathology [174]. Despite these findings, more studies are needed to elucidate the underlying mechanisms of histone methylation changes in AD, as this knowledge could help identify novel therapeutic targets for future drug development.

## 6. Dysregulation of Histone Modification in Parkinson’s Disease

Environmental toxins that induce PD-like conditions are responsible for causing histone acetylation either by CBP upregulation or HDAC downregulation [175,176,177]. For instance, when compared to brain samples from age-matched controls, brain samples from PD patients have higher levels of H3K9ac in the substantia nigra pars compacta [178]. ChIP-seq analysis revealed that PD-related genes such as *SNCA*, *PRKN*, *PARK7*, *MAPT*, and *APP* are present in H3K27 hyperacetylated genome regions. The ChIP-seq study also showed that H3K27 hyperacetylated regions are associated with p300 binding sites. Interestingly, when the ChIPseq data were combined with the RNA-seq data, a negative correlation was found between H3K27ac and transcription in the PD group, but a positive one in the control group, suggesting the decoupling of H3K27ac from the transcription [179]. Another study by Huang et al. found contradictory results. They reported elevated H3K27 acetylation in rotenone-induced dopaminergic N27 cells and even in the substantia nigra of human PD. However, their ChIP-seq and RNA-seq data exhibited that rotenone-induced mitochondrial dysfunction induces H3K27 hyperacetylation, which upregulates transcription and activates neuronal apoptosis [180]. Though a precise explanation for the inconsistencies is still unclear, it is possible that the simultaneous activation of distinct transcription factors leads to the activation of different pathways.

Numerous studies have established the involvement of α-synuclein in histone acetylation and stated that decreased histone acetylation is the cause of the neurotoxicity caused by α-synuclein [168,181]. Furthermore, α-synuclein-mediated toxicity is alleviated by HDAC inhibitors (HDACi) in both cellular and transgenic Drosophila models [178,182]. In dopaminergic neuronal cell lines, the elevated expression of α-synuclein induced considerable transcriptional changes, including a significant downregulation of the DNA repair gene. The overexpression of α-synuclein induces DNA damage that ultimately reduces the histone H3 acetylation level [183,184]. Elevated levels of nuclear α-synuclein lead to reduced histone acetylation, contributing to neurotoxicity [185,186]. Under normal conditions, PGC-1α recruits histone acetyltransferases such as P300 to facilitate histone acetylation and drive the transcription of essential mitochondrial genes, including NRF1/2 and TFAM [187]. However, increased α-synuclein levels interfere with PGC-1α’s function, resulting in decreased histone acetylation and impaired transcription. Specifically, α-synuclein impairs the ability of PGC-1α to regulate mitochondrial biogenesis by disrupting its recruitment to gene promoters [187]. This disruption leads to the diminished expression of mitochondrial biogenesis factors and contributes to mitochondrial dysfunction and cytotoxicity, hallmark features of Parkinson’s disease. Consequently, treatments aimed at enhancing neuronal acetylation may offer promising therapeutic strategies for addressing these pathological processes in Parkinson’s disease [185].

Additionally, an increased nuclear α-synuclein level binds to the gene promoter region of the peroxisome proliferator-activated receptor γ coactivator-1α (PGC-1α) and downregulates PGC-1α expression, concurrently with reduced histone acetylation [188]. α-Synuclein decreases the p300 level and its HAT activity, which may reduce histone acetylation in the dopaminergic neuronal cell line [189].

An elevated level of HDAC2 and a reduced level of Tip60 are seen in the early-stage PD Drosophila model with mutant human α-synuclein (SNCAA30P) [190]. Furthermore, altered binding patterns of Tip60/HDAC2 and a reduced histone acetylation mark such as H4K16ac and H4K12ac in Tip60 target neuroplasticity genes result in the concomitant suppression of these genes. Thus, the deficiencies in short-term memory and locomotion in the PD model are improved by raising the Tip60 HAT levels in the Drosophila brain. These findings suggest that in α -synuclein-linked PD, cognitive deficits are mainly caused by the repression of neuroplasticity-associated genes, which is in turn induced by Tip60-mediated decreased histone acetylation [190].

Dysregulation in histone methylation plays a crucial role in PD pathology by regulating the expression of the SNCA gene and α-synuclein (α-SYN). The upregulated α-synuclein expression in transgenic Drosophila and inducible SH-SY5Y neuroblastoma cells causes the increased methylation of repressive histone marks such as H3K9me1 and H3K9me2 by the upregulation of G9a expression. H3K9me1/2 accumulation represses the expression of the neural cell adhesion molecule L1 (L1CAM) and the synaptosomal-associated protein (SNAP25) and induces impaired neuronal plasticity [168,181]. The inhibition of euchromatic histone-lysine N-methyltransferase 2 (EHMT2) with UNC0638 restores the expression of these genes, suggesting that α-SYN affects histone methylation via EHMT2 [181]. Moreover, treatment with the histone demethylases inhibitor, GSK-J4, restores the histone methylation in H3K27 me3 and H3K4 me3, both of which are linked to PD pathogenesis, and alleviates the dopaminergic neuronal loss and motor deficit in rats [191]. Increased H3K4 me3 promotes SNCA transcription, while H3K27 me3 was found to represses it in a rat model of PD [191]. In contrast, elevated levels of H3K27me3 were observed in the brains of PD patients [192]. H3K4me3 was notably enriched in the SNCA promoter region in postmortem brain samples from both PD patients and matched controls [192]. Similarly, using a dead Cas9-Suntag system for precise locus-specific modifications, reducing H3K4me3 levels at the SNCA promoter resulted in lower α-synuclein expression in neuronal cell lines and PD-derived induced pluripotent stem cells (iPSCs) [192].

## 7. Dysregulation of Histone Modification in Amyotrophic Lateral Sclerosis

Histone acetylation has been found to be related with ALS, as reported by Chen et al., 2017, where the global histone acetylation level was characterized in a yeast ALS proteinopathy model [193]. The results showed that the histone acetylation marks, including H3K14ac and H3K56ac, were significantly decreased in a human FUS-overexpressing yeast model [193]. H3K14ac and H3K56ac are both involved in DNA damage checkpoint activation [194,195]. Furthermore, yeast-overexpressing *FUS* had lower total RNA levels, implying that decreased histone acetylation leads to lower transcription [193]. Interestingly, the same study also used a comparable TDP-43 yeast proteinopathy model and showed hyperacetylation on Lysine 12 and 16 on Histone 4 (H4K12 and H4K16) rather than declines in histone acetylation on H3 [193]. H4K12ac is a modification present in gene promoters that is linked to gene activation [196]; however, H4K16ac is an especially intriguing modification because it is related to both gene expression and repression [197]. Surprisingly, both FUS- and TDP-43-overexpressing ALS models exhibit protein aggregation and cytotoxicity, and their histone modification pattern is different. This suggest that each proteinopathy has its own individual histone modification profile, which may contribute to ALS pathology in different ways.

In the context of histone acetylation, the role of HDACs has also been thoroughly implicated in ALS [198]. The deletion of the Set3-a component of a histone deacetylase complex and homolog to the human protein ASH1, for example, was reported to reduce the toxicity of TDP-43 inclusions in a yeast model [199]. Furthermore, a post-mortem study of ALS patients (brains and spinal cord tissue) revealed a decrease in HDAC1 mRNA and an increase in HDAC2 mRNA [200]. Furthermore, HDAC mislocalization has been linked to ALS disease; for example, in a *FUS* knock-in mouse model, HDAC1 was reported to mislocalize to the cytoplasm [201]. Interestingly, the post-translational modification of the histone modifiers themselves appears to be related with their subcellular localization. HDAC1 phosphorylation in serine residues regulates subcellular distribution, and the nuclear accumulation of HDAC1 found neuroprotective effects in a mouse model [202]. HDAC6 knockdown enhanced SOD1 aggregation, which led to greater motor neuron loss in NSC34 and HEK293 cells as well as in mice expressing wild-type and mutant *SOD1G93A* [203].

A recent study found higher levels of the trimethylation histone marks H3K9me3, H3K27me3, and H4K20me3 around the dipeptide repeat expansions (DRE) in the *C9orf72* gene in brain tissue from ALS patients compared to healthy controls [204]. These histone PTMs are strongly associated with gene silencing and result in a significant reduction in the mRNA level of *C9orf72*. This study suggests that the trimethylation of lysine residues on H3 and H4 is associated with loss-of-function toxicity in ALS patients. In addition, the study demonstrated that using a histone demethylating chemical to treat fibroblasts generated from patients with *C9orf72* mutations resulted in lower tri-methylation levels close to DREs and rescued the *C9orf72* transcription [204].

In yeast models of ALS, the histone methylation profiles for TDP-43- and FUS-induced proteinopathy were found to be unique [193]. Precisely, FUS overexpression corresponds to lower levels of asymmetric di-methylation on Arginine 3 in Histone H4 (H4R3me2asym), whereas TDP-43 overexpression is linked to lower levels of the H3K36me3 mark. Furthermore, H34Rme2asym has been connected to increased transcription [205], while H3K36me3 is related to transcriptional suppression by acting as a binding site for HDACs which promotes deacetylation [206]. Thus, similar to acetylation, different proteinopathies have distinct methylation patterns that contribute to the disease pathophysiology in their own way.

Protein arginine N-methyltransferase 1 (PRMT1) is a methyltransferase that has been linked to ALS. PRMT1 is responsible for H4R3me2asym, which increases histone acetylation and gene expression [207]. In a FUS R521C mouse model of ALS, the PRMT1 activity was downregulated by an interaction with FUS, making a stable complex of FUS/PRMT1/Nd1-L mRNA. The same study also showed that the overexpression of PRMT1 was found to rescue neurite growth after oxidative stress [208]. Additionally, the loss of PRMT1 function, caused by FUS mislocalization, led to a reduction in the asymmetric di-methylation of Arginine 3 on Histone H4 (H4R3me2asym), which in turn caused a drop in the acetylation of Histone H3 on Lysine 9 and 14, ultimately leading to transcriptional silencing [209].

Therefore, the above discussion clearly shows that most cases of AD, PD, and ALS are sporadic and brought on by intricate gene–environment interactions during the course of life. Indeed, animal studies showed that early life environmental stressors including chemical exposure, nutritional limits, maternal stress, and so on can affect the neuronal epigenome, resulting in neurocognitive, neurobehavioral, and/or neurodegenerative problems later in life [210].

## 8. Therapeutics

Since there has been a continuous surge in NDDs, potential therapeutic drugs are utterly needed. The available medications used to treat NDDs act only to manage the disease symptoms. For instance, Cholinesterase inhibitors to treat AD or L-dopa for PD replace neurotransmitters and improve disease symptoms but have no subtle effect on the disease progression pathway. Moreover, there are only a limited number of drugs available to treat AD, PD, and ALS (Table 1). These drugs have certain limitations and impart side-effects if taken for too long. So, there is an urgent need to look for alternative therapeutic options to treat NDDs. Recent developments in neuroepigenetics are opening new doors for drug development research that target the disease progression pathway and alleviate disease symptoms. In this section, we highlight the potential power of new therapeutic approaches that target the epigenetic machinery.

### Epigenetic Therapy for Common Neurodegenerative Disorders

Over the last decade, novel chromatin-modifying medications, also referred to as ‘epidrugs’, have been developed and advanced to clinical trials for treatment in patients with neurological conditions. To date, the most promising treatments are HDAC inhibitors and DNA-demethylating agents. Members of both categories have been used to treat cancer for about two decades and have been licensed by the US Food and Drug Administration for hematological malignancies. With the approval of epidrugs targeting DNMT (azactidine and decitabine) and HDACs, namely suberoylanilide hydroxamic acid (SAHA) and romidepsin, epigenetic-based therapeutics have made considerable strides [211].

In psychiatry and neurology, HDAC inhibitors have long been used as anti-epileptics and mood stabilizers. They have recently been studied as prospective therapeutic agents for the treatment of neurological disorders because of their anti-inflammatory properties [212]. The evidence from preclinical research suggests that HDAC inhibitors may have therapeutic value in neurological conditions such as AD, PD, acute brain injury, stroke, and Huntington disease [212]. HDAC inhibitors can be grouped into four major chemical families: namely, (1) hydroxamic acids like Trichostatin A and SAHA, (2) epoxyketones such as trapoxin, (3) short-chain fatty acids such as sodium butyrate, phenylbutyrate, and valproic acid (VPA), and (4) benzamides [213]. Butyrates stand out among them because of their propensity to cross the blood–brain barrier, making them potentially effective for treating brain diseases.

Histone deacetylase (HDAC) inhibitors play a significant role in enhancing neuronal survival, motor function recovery, and neuroplasticity, especially in stroke conditions. Inhibiting HDAC2 using non-selective inhibitors such as MGCD0103, SAHA, and TMP269 was found to promote neuronal survival, suppress neuroinflammation, and improve motor function post-stroke, highlighting the critical role of HDAC2 in recovery [214]. Other HDAC isoform inhibitors were ineffective. Further studies confirmed that HDAC2, but not HDAC3 or HDAC1, played a key role in PTS-induced injury, as selective HDAC2 inhibitors reduced apoptosis and infarct size, aiding recovery. The kinetically selective HDAC2 inhibitors BRD6688 and BRD4884 significantly enhanced the acetylation of histones H4K12 and H3K9 in primary mouse neuronal cell cultures [215]. Selective HDAC3 inhibition (via RGFP966) also showed protective effects by reducing the infarction size, modulating inflammatory responses, and alleviating ischemic cerebral injury [216]. RGFP966 attenuated STAT1 phosphorylation and AIM2 inflammasome activation, offering neuroprotection. Similarly, selective HDAC8 inhibitors like PCI-34051 exhibited neuroprotective properties, although their mechanism was likely unrelated to direct HDAC8 inhibition and instead involved antioxidant activity by metal binding [216].

The inhibition of class IIa HDACs by MC1568 showed mixed outcomes, worsening brain recovery in some studies through CREB and c-Fos inactivation [217], while reducing the infarct volume and neurological deficits by activating NCX3 transcription in other studies [218]. HDAC6 inhibitors, like tubastatin A and HPOB, significantly reduced ischemia-induced apoptosis, decreased the infarct volume, promoted axon regeneration, and restored key proteins like α-tubulin and FGF-21 [219,220]. HDAC6 inhibition also improved the endothelial function and reduced ischemic brain damage in other models. Newly developed HDAC6 inhibitors, ACY-738 and ACY-775, are bioavailable in the brain and show promise in treating peripheral neuropathy and depression [221].

Furthermore, SIRT1, a NAD-dependent HDAC implicated in the neurodegeneration associated with AD and HD, has piqued the curiosity of many researchers [222,223]. In accordance with this, the development of new medications that inhibit nicotinamide binding to SIRT1 and other NAD-dependent HDACs should pave the way for the emergence of a new generation of drugs that target the epigenetic machinery with more specificity. However, the use of these medications to treat neurological conditions is still in its early stages.

To control DNA methylation, two treatment approaches can be used: the first involves the use of DNMT inhibitors, and the second involves the administration of methyl donor substances such as folates and other B-group vitamins needed for SAM formation [224]. DNMT inhibitors include azacitidine and decitabine; both are FDA-approved drugs for the treatment of hematological malignancies. Unfortunately, these chemicals are frequently toxic and unstable; they appear to work better against rapidly reproducing cancer cells than in non-proliferating neurons. For example, decitabine has been found to accelerate neurotoxicity and upregulate PD-related genes in cultured dopaminergic neurons, including the demethylation of the *SNCA* gene, which codes for α-synuclein [225]. SAM is a ubiquitous intracellular methyl donor molecule produced by a one-carbon metabolism, which is an important metabolic route that connects the folate and methionine cycles. Therefore, dietary folates and related B-group vitamins are, therefore, essential for SAM synthesis, and their deficiency can impede cellular methylation capability [226]. Recently, it was shown that *PSEN1* hypomethylation led to increased gene expression in the blood of AD patients as well as post-mortem AD brain regions, implying that it might be an epigenetic biomarker for the disease [227]. This finding was inconsistent with previous research that reported the diet low in the methyl donor compound aggravate amyloid-beta production by the hypomethylation of *PSEN1* [228]. This study also reported that the dietary supplementation of SAM restored *PSEN1* methylation levels and improved behavioral deficits in animals, suggesting that it has therapeutic potential for AD [228]. The research reported SAM as a potential treatment for AD, highlighting its epigenetic and antioxidant properties [224]. In animal and cell culture models, SAM reduced amyloid-beta accumulation and boosted antioxidant defenses. In human trials, formulations combining SAM with folic acid, vitamin B12, and nutraceuticals such as α-tocopherol, N-acetyl cysteine, and acetyl-L-carnitine showed some cognitive improvements in AD patients [224]. However, these trials did not assess DNA methylation changes, leaving the epigenetic effects of these formulations in humans unconfirmed [224].

Research found that inhibiting the H3K4-specific methyltransferases with the compound WDR5-0103 significantly restored the synaptic function and improved memory-related behaviors in AD mice [170]. Among the genes upregulated in the prefrontal cortex of AD mice, many showed increased H3K4me3 enrichment in their promoters, which was reversed by treatment with WDR5-0103 [170]. One of the top target genes, *Sgk1* (which encodes serum and glucocorticoid-regulated kinase 1), was also significantly elevated in the PFC of AD patients. The inhibition of *Sgk1* by WDR5-0103 treatment reduced hyperphosphorylated tau, restored glutamatergic synaptic function, and improved memory deficits in AD mice, highlighting its potential as a therapeutic target in AD [170].

The research highlights that while DNMT3a/b are primarily responsible for de novo DNA methylation and DNMT1 for maintaining it, their roles overlap. Both DNMT1 and DNMT3a need to be knocked out in adult forebrain neurons to induce impairments in long-term synaptic plasticity and learning and memory deficits [229]. Additionally, studies using a Tet1 knockout mouse model and RNA knockdown experiments show that the Tet1-mediated oxidation of methylcytosine (mC) is crucial for memory and regulating genes involved in neuronal activity, such as *Fos* and *Arc*, particularly in the dorsal hippocampus [230].

Researchers explored the potential role of 5-aza-2′-deoxycytidine (5-aza-dC), a DNMT inhibitor, for treating PD. They found that 5-aza-dC induced CpG demethylation in the promoter region of the *α-synuclein* gene, leading to its upregulation [225]. Additionally, 5-aza-dC increased tyrosine hydroxylase expression, enhancing dopamine production, while also elevating *α*-synuclein levels [225]. If levodopa, a common Parkinson’s treatment, functions through epigenetic mechanisms, existing therapies should be reassessed to uncover new epigenetic pathways and guide the development of more targeted drugs [231]. Methylation-related compounds like vitamin B, folic acid, and SAMe are undergoing clinical trials for neurodegenerative diseases as folate and vitamin B6 are commonly prescribed to reduce homocysteine, a risk factor for Alzheimer’s disease [232]. Similarly, DNMT inhibitors such as 5-aza-cytidine (5-azaC), 5-aza-2-deoxycytidine (decitabine), zebularine, and RG108 have been shown to improve outcomes in ALS. Specifically, RG108 inhibits DNA methylation in motor neurons, contributing to disease improvement [233].

In PD animal models, epigenetic therapies focused primarily on histone tail modifications rather than DNA methylation [234]. Based on a recent literature review on the use of HDAC inhibitors for the treatment of AD [224], PD [234], ALS [235], and other NDD in animal models, it was found that HDAC inhibitors (HDACi) were able to enhance synaptic plasticity as well as cognitive and motor functions in the animals. However, most of these chemicals are neurotoxic and nonspecific, raising concerns about long-term human treatment. For instance, the HDACi VPA is an anti-epileptic medicine used as a first-line treatment for the majority of epilepsy cases; nevertheless, the long-term use of VPA in people led to Parkinsonism, despite the fact that VPA treatment was neuroprotective in PD animal models [236]. Additionally, there is currently no agreement among animal studies regarding the specific HDAC isoforms that should be selectively inhibited to achieve the therapeutic effect and an adequate safety profile in neurons, so HDACi clinical trials are ongoing to assess their safety and efficacy in patients with NDD [237].

To assess the effectiveness of synergistic effects on the multitarget drug approach, approved and experimental epidrugs are being tested in combination with standard medicines in the field of cancer research. The research highlights the effectiveness of epigenetic therapy in treating refractory metastatic non-small cell lung cancer (NSCLC). The combination of 5-azacytidine and entinostat proved as effective as conventional chemotherapy with erlotinib [238]. A promising example of this synergistic approach comes from a recent phase II study, which combined 5-azacytidine with the thrombopoietin mimetic romiplostim in patients with myelodysplastic syndrome (MDS), showing positive outcomes [239]. Additionally, any potential combination with erlotinib should consider the recently identified link between cancer cell resistance to this drug and the increased expression of the histone demethylase KDM5, which could affect treatment efficacy [240].The study also highlights the potential of monitoring DNA methylation changes in poor prognosis biomarkers (e.g., CDKN2A, CDH13, APC, and RASSF1A) in free-circulating tumor DNA, as demethylation correlated with positive responses. This and other markers, such as DNMT3B amplification, could serve as predictors of a clinical benefit in future trials involving DNA methyltransferase inhibitors (DNMTis) [241].

A similar strategy can be anticipated to treat complicated disorders like neurodegeneration. However, the basic mechanism of disease progression in both diseases is quite different. In cancer, cells proliferate rapidly, necessitating the continuous rewriting of epigenetic marks in daughter cells, so that inhibiting epigenetic machinery enzymes with DNMT or HDAC inhibitors can reactivate genes whose silencing is responsible for cancer aggressiveness and resistance to conventional therapies. On the other hand, neuronal death is a hallmark of NDD, and most of their signs and symptoms only appear after a significant number of neurons have already died. As a result, treating individuals who are already suffering may be too late, and boosting the plasticity of the surviving neurons with epigenetic agents may only delay disease progression without considerably improving quality of life. For instance, prolonged VPA treatment did not improve mortality or the course of the disease in ALS patients, despite promising outcomes in transgenic ALS mice [242].

Overall, based on in vitro and in vivo research on neurodegeneration, targeting epigenetic markers may be a promising approach for improving memory and cognitive function in individuals suffering from NDD. Despite some promising results, most of the epidrugs have been associated with significant negative consequences [243]. Bio-based natural products called nutraceuticals, however, have various benefits over conventional medications, including a lower incidence of adverse effects, cost-effectiveness, widespread availability, a synergistic action mechanism, and the ability to enhance overall patient health [244]. Together, these properties make nutraceuticals a viable alternative to traditional medicine for addressing a wide range of medical conditions. Dietary supplements have numerous biological effects, some of which are mediated through epigenetically controlled gene expression. Natural substances such as polyphenols, isoflavones, isothiocyanates, γ- and α-tocopherols, and a number of vitamins (B6, B9, and B12) have been shown to have epigenetic effects against NDDs in recent studies [245,246]. In this section, we have endeavored to illuminate the potential of nutraceuticals as an innovative therapeutic strategy aimed at modulating the epigenetic machinery (Table 2).

Curcumin, a phenolic compound found in turmeric, is used to treat dyspepsia, stress, and mood disorders [259]. It specifically inhibits the HAT activity of the protein p300/CREB-binding protein (CBP) [260]. Curcumin is a highly effective HDAC inhibitor, surpassing the well-studied HDAC inhibitors valproic acid and sodium butyrate [261]. A formulation of Curcumin (Theracurmin, 90–180 mg/day, for 18 months), having p300/CBP HAT inhibitory activity, is currently in phase II clinical trials to test its efficacy to improve cognitive function and reduce Aβ-deposition in AD patients (ClinicalTrials.gov Identifier: NCT01383161, accessed on 29 October, 2024) [251,262]. Curcumin significantly attenuates H3 acetylation in the promoter region of AD-related genes PS1 and BACE1 and downregulates their expression by inhibiting the HAT activity of p300 [250]. Vitamin B plays a considerable role in neurogenesis and the effect of vit B treatment on the brain is significant. Vitamin B12 is an effective supplement in the pharmaceutical cocktail used to treat NDD and its deficiency has been associated with aberrant memory, poor cognition, and AD [261]. PSEN1 hypomethylation increases PSEN1, BACE1, and APP protein levels and induces Aβ deposition in the mouse brain [261]. Treatment with vitamin B12 restores DNA methylation patterns and reduces the Aβ burden [263]. In AD patients as well as in a rodent model, supplementing the diet with folate, vitamins B6, and B12 improves memory impairment by reducing tau hyperphosphorylation and NFT deposition in the hippocampus and cortex [264]. A vitamin B12 deficiency affects the activity of DNMT1, DNMT3a, and DNMT3b [265].

Resveratrol is a polyphenolic compound present in berries, grains, grapes, seeds, tea, and vegetables, etc. It is a strong antioxidant with anti-inflammatory qualities and is effective in preventing cancer and brain disorders [266]. In animal models of neurodegeneration, resveratrol and its derivatives act as SIRT activators. In a primary neuronal culture of rat cortices, resveratrol was found protective against Aβ-toxicity by the upregulation of SIRT1 and the suppression of NF-kB signaling [267]. The neuroprotective function of resveratrol is partially mediated by the SIRT1-serine/threonine protein kinase ROCK1 [268] and SIRT1-Akt pathways [269], which increase β-secretase activity. Thus, resveratrol treatment ameliorates hippocampal damage and Aβ plaque formation in mouse models of AD and improves brain function [270]. PD pathogenesis has been linked to mitochondrial dysfunction for more than two decades. In cultured primary fibroblasts from patients with familial PD, which is associated with different Park2 mutations, resveratrol improves the mitochondrial respiratory capacity by activating the AMPK-SIRT1 signaling pathway and inducing PGC-1 activity [247]. PGC-1 activation improves mitochondrial biogenesis and increases the mitochondrial oxidative function, both of which are possible targets for PD therapy.

Epigallocatechin gallate (EGCG) is the main polyphenolic compound present in green tea. A significant study has been completed on the neuroprotective properties of EGCG and its metabolites, and a positive correlation was found between regular tea consumption and enhanced cognitive function [271].

E-PodoFavalin-15999 (AtreMorine) is a new biopharmaceutical molecule extracted from the Vicia faba L. plant using a non-denaturing biotechnological technique known as ultrapure lyophilization. The E-PodoFavalin-15999 extract is a natural source of L-DOPA and acts as the structural base of AtreMorine [272]. L-DOPA has been extensively studied for its effect on DNA methylation, and found to upregulate the level of different proteins involved in methylation [273]. Furthermore, research has shown that AtreMorine reverses the alterations in DNA methylation that occur in NDDs such as AD and PD, returning levels similar to the controls [274]. AtreMorine was found to be neuroprotective in a triple-transgenic mouse model of AD by upregulating DNMT3a mRNA expression and increasing global DNA methylation while lowering HDAC3 expression. Additionally, the control of DNA methylation by AtreMorine is unrelated to dopamine levels or pathways, indicating that both dopamine-dependent and -independent processes are involved in its neuroprotective effects [257].

Another novel compound called nosustrophine is a porcine-derived bioproduct prepared using the same method of ultrapure lyophilization as Atremorine. This technique preserves the structural and functional integrity of the biologically active components in nosustrophine. Nosustrophine contains several nutritional supplements including glutamate; aspartate; norepinephrine/noradrenaline; dopamine; serotonin; BDNF; neurotrophic tyrosine kinase receptor type 3 (NTRK3/TrkC); neuropeptide Y; vitamins B2, B3, D3, and E; calcium; iron; magnesium; zinc; among others. Nosustrophine targets the epigenetic machinery by altering DNA methylation as well as SIRT and HDAC expression and controls AD-linked gene expression [258].

## 9. Conclusions and Future Perspectives

The discussion so far has concluded that the NDD discussed in this review, including AD, PD, and ALS, share several epidemiologic and genetic features. First, they all have an etiologic dichotomy, with more common late-onset forms and less common familial forms. It is possible (and likely) that a significant proportion of cases that were previously thought to be sporadic and nonfamilial will ultimately be found to result from disease-causing mutations or genetic risk factors (like APOE-ε4 in AD). Second, in some instances, identical mutations and polymorphisms have been linked to and associated with a variety of clinically and neuropathologically distinct disease entities. GWAS has a significant role in this effort because they have enabled the discovery of novel genetic associations that are not based on prior knowledge. Unfortunately, GWAS findings, so far, have explained only a small proportion of the heritability of complex diseases, making genetic risk prediction tests for these diseases currently unfeasible.

The majority of NDD are sporadic in nature and develop over time with gene–environment interactions, which causes chromatin modification and disease progression. The alterations in the chromatin structure and function that take place in NDD and how they affect the pathophysiology of the disease are still poorly understood. However, new findings emphasize the significance of maintaining chromatin dynamics and adequate amounts of histone and DNA modifications, with imbalances resulting in deteriorating effects. All neurological conditions are age-dependent, which raises the possibility that these modifications could build up over time and eventually collapse in order to cause irreversible damage to the neurons. An earlier treatment with the epigenetic drugs, that attempted to restore the correct gene expression and chromatin dynamics, may offer novel therapeutic approaches. The epigenetic drugs could be given in combination with the treatments for other symptoms of these disorders, such as protein aggregation and misfolding.

In summary, the development of novel, more stable, selective, and less toxic epigenetic drugs could be the promising way to improve neuronal function and delay disease progression in individuals already suffering from NDD. However, we must shed light on the complex gene–environment interactions that occur throughout life and use recent genome-wide technologies to identify epigenetic biomarkers capable of detecting individuals at risk of developing neurodegeneration later in life. This information could be beneficial in epigenetic therapies using natural substances, exercise, brain stimulation, and other methods that could increase neural plasticity to stop or delay the onset of disease.

## Figures and Tables

**Figure 1 ijms-25-11658-f001:**
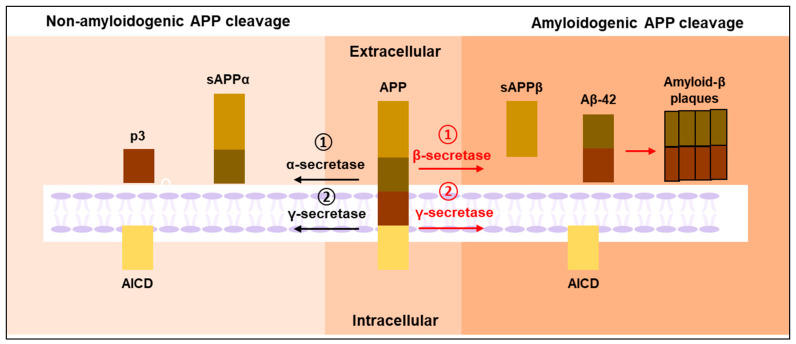
APP processing in normal physiological state and disease condition. In the amyloidogenic pathway, APP undergoes processing by β-secretase, leading to the release of sAPPβ and the formation of a membrane-bound fragment, which is then cleaved by γ-secretase, resulting in the production of Aβ and AICD. Conversely, in the non-amyloidogenic pathway, APP is cleaved by α-secretase, yielding sAPPα and a membrane-bound fragment, and subsequent cleavage by γ-secretase results in the generation of p3 and AICD.

**Figure 2 ijms-25-11658-f002:**
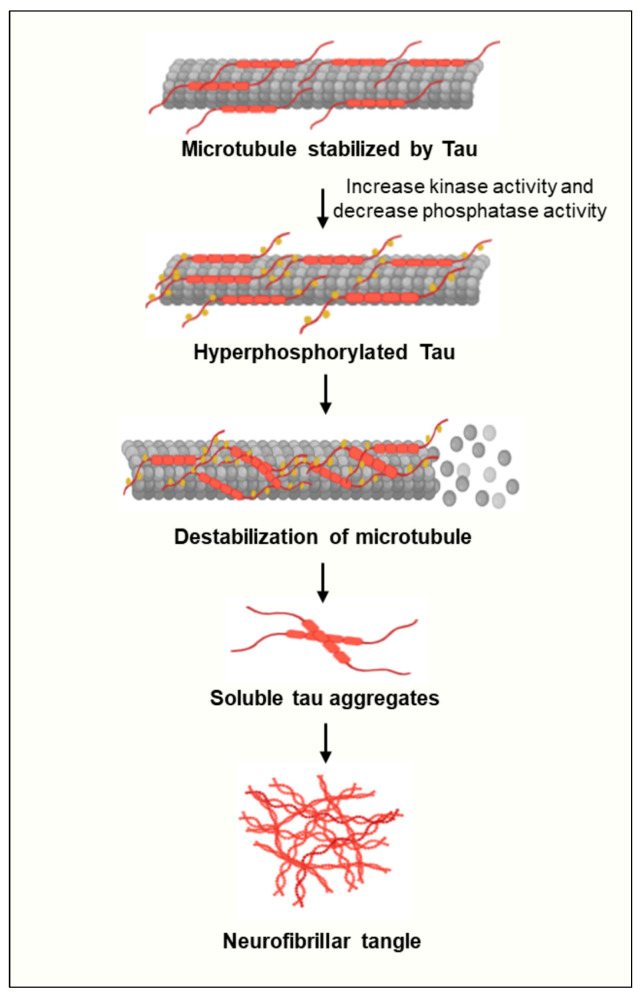
Schematic representation of the process leading to neurofibrillar tangle (NFT) formation in Alzheimer’s disease. Under physiological conditions, tau functions as a microtubule-associated protein. Pathological tau, prone to aggregation, undergoes hyper-phosphorylation, resulting in destabilization and dissociation of microtubules. Subsequently, soluble phosphorylated tau molecules aggregate to form NFTs. NFTs, Neurofibrillary tangles.

**Figure 3 ijms-25-11658-f003:**
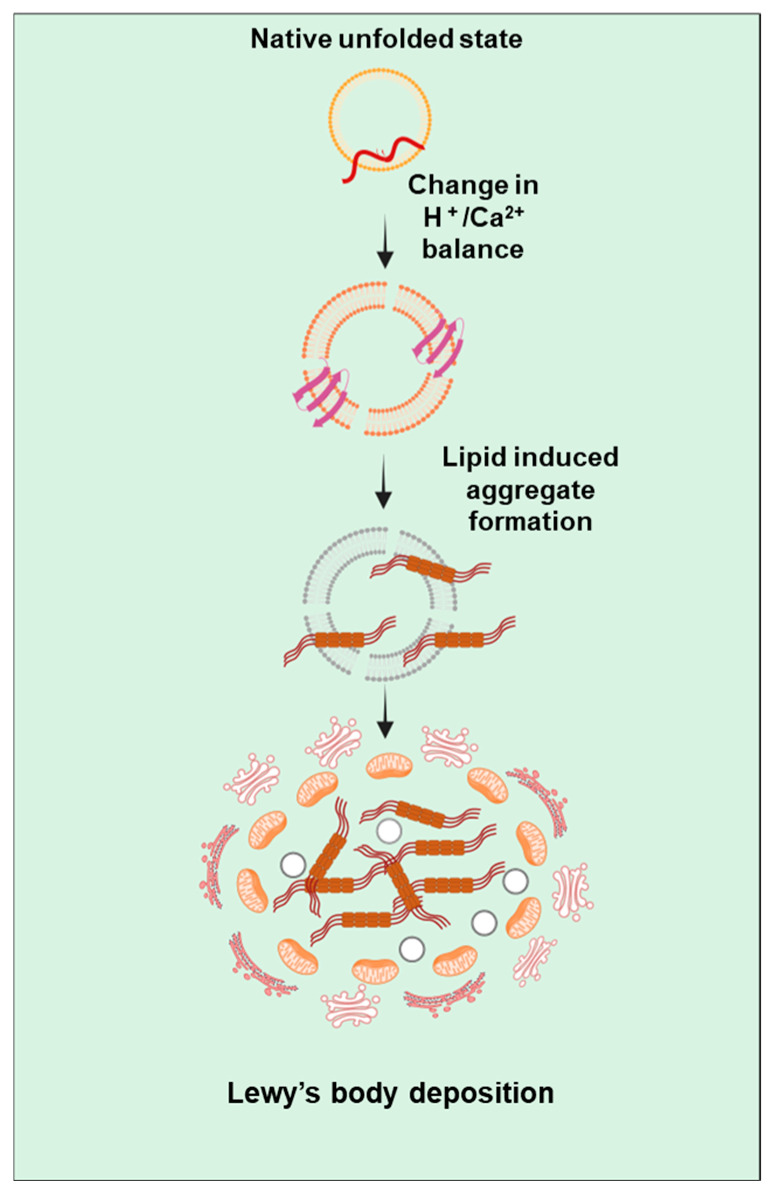
Illustration depicting the pathway of α-synuclein aggregation. Under pathological conditions, α-synuclein aggregation can occur either in association with the cellular membrane or within the cytosol. When bound to the membrane, monomeric α-synuclein adopts an α-helical conformation; however, ionic dysregulation prompts a conformational shift towards membrane-bound β-sheet structures, leading to self-association and the formation of oligomers and fibrils. The haphazard accumulation of these fibrils contributes to the formation of intracytoplasmic Lewy bodies. Throughout α-synuclein fibrillogenesis, oligomers and amyloid fibrils exert significant toxicity, impairing microtubule dynamics, endoplasmic reticulum–Golgi trafficking, and mitochondrial function.

**Figure 4 ijms-25-11658-f004:**
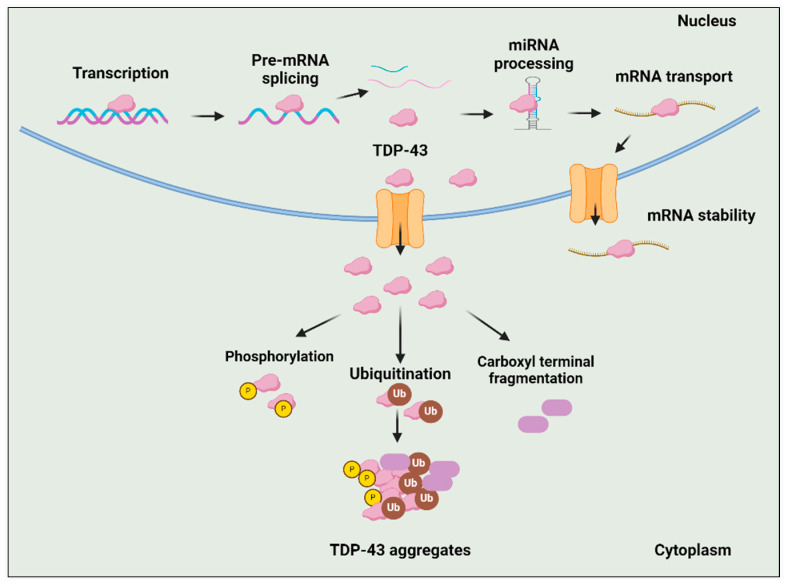
Function of TDP-43 in Normal and Disease Conditions. TDP-43 plays diverse functional roles including initiation of transcription, pre-mRNA splicing, miRNA processing, mRNA transport, and mRNA stability. However, under pathological conditions, TDP-43 is depleted from the nucleus and aggregates in the cytoplasm in hyperphosphorylated, ubiquitinated, and cleaved forms (CTFs), which are characteristic features of amyotrophic lateral sclerosis (ALS) and frontotemporal lobar degeneration (FTLD) proteinopathies.

**Figure 5 ijms-25-11658-f005:**
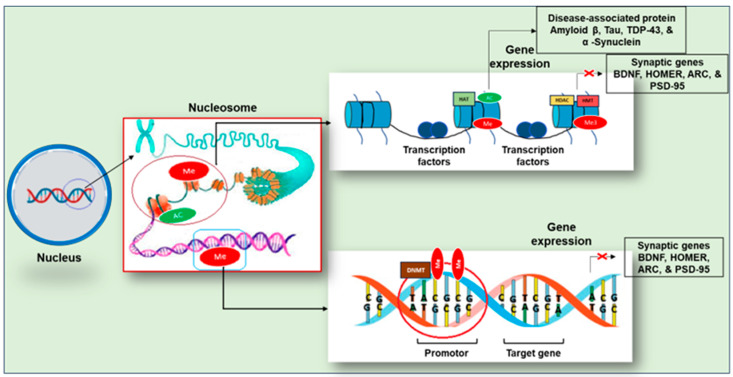
A general outline of epigenetic modification in neurodegeneration. Histone acetylation mediated by HAT is associated with gene expression, while histone methylation is mediated by HMT; monomethylation of histone is associated with gene expression whereas trimethylation causes gene repression. DNA methylation is mediated by DNA methyltransferase (DNMT) which is also related with gene repression. Abbreviation: Me—methylation; Ac—acetylation, Histone acetyltransferase (HAT), histone methyltransferase (HMT). The red crosses show inhibition of gene expression.

**Figure 6 ijms-25-11658-f006:**
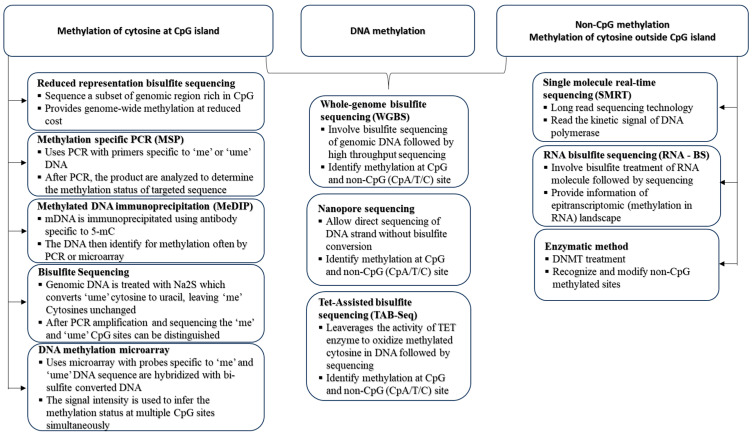
Visualization of diverse methodologies utilized in the analysis of DNA methylation, covering CpG and non-CpG regions. Method selection considerations include assessing the desired resolution, specificity, and coverage for a thorough examination of DNA methylation patterns, especially within both known and unknown genomic contexts. Abbreviation: me—methylated, unme—unmethylated.

**Figure 7 ijms-25-11658-f007:**
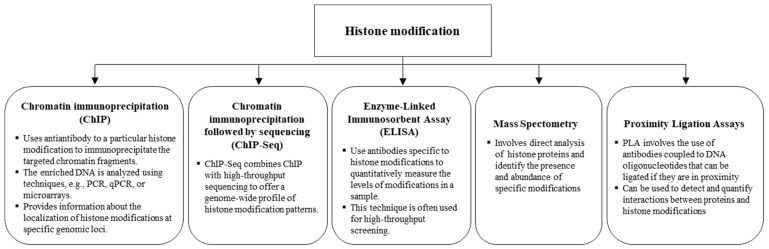
Methods for analyzing histone modification, including Chromatin Immunoprecipitation (ChIP), ChIP-sequencing (ChIP-seq), Enzyme-Linked Immunosorbent Assay (ELISA), Mass Spectrometry (MS), and Proximity Ligation Assay (PLA). Consideration for method selection depends on the desired scale, specificity, and quantitative precision of histone modification analysis.

**Table 1 ijms-25-11658-t001:** List of FDA-approved drugs to treat ALS, AD, and PD.

Disease	Drug Name	Approved in	Nature	Mechanism
ALS	Qalsody/Toferson	2023	Antisense oligonucleotide	Binds to the RNA produced from mutated SOD1 gene and stop SOD1 protein formation.
	AMX0035 (Relyvrio)	2022	A combination of sodium phenylbutyrate (PB) and taurursodiol (TURSO)	Target both mitochondria and ER and prevent motor neuronal death.
	Radicava™ (edaravone)	2017	Free radical scavenger	Scavenges ROS and protects neurons.
	Rilutek (Riluzole)	1995	Benzothiazole class	Inhibit glutamate release (mechanism unknown).
AD	Galantamine (Razadyne^®^)	2001	Cholinesterase inhibitor	Inhibit Cholinesterase activity and increase Acetylcholine concentration.
	Rivastigmine (Exelon^®^)	2007		
	Donepezil (Aricept^®^)	1996		
	Lecanemab (Lequembi^®^)	2023	Humanized immunoglobulin gamma 1 (IgG1) monoclonal antibody	Directed against aggregated soluble and insoluble forms of amyloid beta and reduces Aβ burden.
	Aducanumab (Aduhelm^®^)	2021		
	Memantine (Namenda^®^)	2003	NMDA receptor Antagonist	Restore the function of damaged nerve cells and reduce the abnormal excitatory signal by reducing the glutamate level.
	Levodopa/carbidopa	2015	Levodopa is metabolic precursor of dopamine Carbidopa inhibits decarboxylation of aromatic amino acid	Carbidopa inhibits the decarboxylation of peripheral levodopa, making more levodopa available for delivery to the brain, where it is converted to dopamine and increases dopamine concentration.
PD	Apomorphine	2004	Non-ergoline dopamine agonist	Mimic dopamine.
	Pramipexole	1997		
	Rotigotine	2007		
	Entacapone	1999	Inhibitor of catechol-O-methyltransferase (COMT) which breaks down dopamine	Improve dopaminergic signaling.
	Amantadine	2017	Weak uncompetitive antagonist of the NMDA receptor	Unknown.

**Table 2 ijms-25-11658-t002:** List of potential of nutraceuticals that target the epigenetic machinery in neurodegenerative disorders.

Name	Epigenetic Factor	Disease	Mechanism	Effect	Clinical Trial
**Resveratrol**	SIRT1 Activator	PD Mitochondrial dysfunction Abnormal protein deposition (α-synuclein) AD▪Tau phosphorylation and NFT deposition	AMPK-SIRT1 signaling pathway, inducing PGC-1 activity. SIRT1 deacetylates heat shock factor 1 and increases the transcription of heat shock proteins (Hsp70s). SIRT 1 deacetylates p53, reducing its expression, disturbing p53-GSK3β interaction.	Improves mitochondrial function [247]. Prevents the production of pathological protein aggregates [248]. Reduces tau phosphorylation and NFT deposition [249].	
**Curcumin** **Theracurmin (90–180 mg/day)**	p300/CBPHAT Inhibitor	AD▪*PS1*▪*BACE 1*▪Aβ plaque deposition	Inhibiting H3 acetylation at the promoter of gene by inhibiting p300 HAT.	Supress *PS1* and *BACE 1* expression [250]. Reducing Aβ accumulation [251].	Phase II clinical trial (ClinicalTrials.gov Identifier: NCT01383161) [251]
**Folic acid** **Vit B9****Vit B6****Vit B12**	Increase SAM	AD ADPDPD	Upregulate Methylation of Aβ-40, PS1.	Reduces Aβ burden [252]. Downregulates cytokine expression [253].Improves cognitive dysfunction [254].	(ChiCTR-TRC-13003246) [252]
**EGCG**	DNMT inhibitor HDAC inhibitor	AD SAMP8 mouse model of AD	Alters DNA methylation profile. Decreases HDAC activity, increases H3 and H4 acetylation, and downregulates HDAC1, HDAC2, and G9a expression, respectively.	Prevention of Aβ aggregation in AD [255]. Facilitates the degradation of Aβ [256].	Phase IIand III clinical trials(NTC00951834) [232]
**AtreMorine**	DNMT activator	Triple transgenic mouse model of AD	Upregulates DNMT3a and increases global DNA methylation.	Reduces plaque formation [257].	
**Nosustrophine**	SIRT1 activator	AD	Deacetylation of H3K14.	Promotes survival by attenuating Bax-dependent apoptosis [258].

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
