# Peer review of "Epigenetic Explorations of Neurological Disorders, the Identification Methods, and Therapeutic Avenues"

_ijms, 2024, doi:10.3390/ijms252111658_

Round 1
Reviewer 1 Report
Comments and Suggestions for Authors
The authors wrote a review about the burgeoning realm of neuroepigenetics, emphasizing its role in enhancing our mechanistic comprehension of neurodegenerative disorders and elucidating the predominant techniques employed for detecting modifications in the epigenome.
Comments:
The article is well organized.
Introduction: as authors already said, all classical NDD are included (AD, PD, ALS, FTD and HD). However, multiple sclerosis is one of the important NDDs as well and shouldn't be excluded.
The references are updated, all figures and tables are nicely presented. However, make sure that figure or table legends include all abbreviations explanations as well.
Author Response
: Thanks for the comments. Huntington's disease was not included in this review primarily due to the authors' intention to focus on neurodegenerative disorders that are more commonly addressed together in the literature, allowing for a more targeted exploration of the shared mechanisms and therapeutic strategies among Alzheimer's disease, Parkinson's disease, and amyotrophic lateral sclerosis. While recognizing the overlapping pathophysiological features of Huntington's disease, the authors felt that a focused analysis of the selected disorders would provide deeper insights into their unique and interrelated biological processes. In addition, we include all abbreviations in figure and table legends as suggested
Reviewer 2 Report
Comments and Suggestions for Authors
Review of the article titled "Epigenetic Explorations of Neurological Disorders, the Identification Methods, and Therapeutic Avenues".
Improvement Suggestions:
1. Introduction to such a broad scientific area is very scarce. Epidemiology is larger and mainly oriented to USA data, which does not reflect the international characteristics of IJMS. Enhance the discussion on neuroinflammation and epigenetic regulation (Lines 35-37). Neuroinflammation is a common feature of NDDs but is not linked explicitly to epigenetic regulation. Discuss how epigenetic changes (e.g., in cytokine gene promoters) can drive neuroinflammatory processes in diseases like Alzheimer’s and ALS. Provide examples of critical pathways and potential epigenetic targets for modulating neuroinflammation.
2. Clarification of tau phosphorylation’s role in neurodegeneration (Lines 86-104). The section on tau pathology in Alzheimer’s disease does not sufficiently emphasize the interaction between tau hyperphosphorylation and other cellular processes like autophagy. Discuss the link between tau phosphorylation and autophagy dysfunction, highlighting key kinases involved (e.g., GSK3β) and their potential as therapeutic targets.
3. Add depth to the discussion on mitochondrial epigenetics in NDDs (Lines 125-132). Mitochondrial dysfunction is mentioned but not linked to specific epigenetic modifications. Include studies that explore the connection between mitochondrial DNA methylation, histone modifications, and neurodegeneration, especially in diseases like Parkinson's and ALS.
4. Expand discussion on environmental factors influencing neuroepigenetics (Lines 162-167). The role of environmental factors in modifying epigenetic landscapes is briefly mentioned without specific examples. Provide examples of environmental stressors (e.g., toxin exposure, diet) and their known effects on epigenetic regulation in the brain. Cite studies showing how these factors can contribute to disease onset or progression.
5. The review includes miRNAs and their epigenetic role in NDDs (Lines 184-186). It mentions miRNAs but does not provide detailed examples of their role in neurodegenerative diseases. Include specific examples of miRNAs, such as miR-9 or miR-124, which are known to regulate genes involved in synaptic plasticity and neuronal survival in diseases like Alzheimer’s and Parkinson’s.
6. Clarification of chromatin remodeling enzymes in epigenetic therapy (Lines 207-212). Chromatin remodeling enzymes are mentioned in the context of neuroepigenetics, but their mechanistic role is not clear. Add details about specific enzymes (e.g., SWI/SNF complexes) and their involvement in regulating transcriptional responses to neurodegenerative stimuli. Discuss their potential as therapeutic targets in NDDs.
7. More comprehensive mechanistic details for DNA methylation in neurodegeneration (Lines 245-246). The discussion on the role of DNA methylation in synaptic plasticity and cognition lacks depth. Expand on how specific genes like BDNF, ARC, or EGR1 are involved in learning and memory, integrating more experimental evidence, especially from animal models or human studies.
8. Greater focus on the heterogeneity of histone modifications in NDDs (Lines 266-273). The section on histone methylation does not adequately address the diverse functional outcomes of different modifications. Provide more examples of specific methylation marks (e.g., H3K27 versus H3K9) and their differential roles in transcriptional repression versus activation in neurodegenerative disorders.
9. Address limitations in current single-cell epigenetic profiling technologies (Lines 363-372). The limitations of single-cell bisulfite sequencing are briefly mentioned but could be more thoroughly explored. Discuss the technical challenges in single-cell epigenomics, such as DNA degradation, low coverage, and the difficulty of integrating multi-omics data. Suggest potential solutions or emerging technologies that may overcome these challenges.
10. Highlight recent advances in three-dimensional chromatin structure analysis (Lines 433-450). The section on chromatin organization techniques, such as Hi-C, lacks recent developments and applications. Add recent studies that utilize advanced 3D chromatin mapping techniques to study neurodegeneration, such as how topologically associated domains (TADs) might reorganize in NDDs, affecting gene expression.
11. Clarification on the role of α-synuclein in histone modifications in Parkinson's Disease (Lines 511-521). The impact of α-synuclein on histone acetylation is noted but underdeveloped. Elaborate on the specific molecular pathways through which α-synuclein interferes with histone acetylation, particularly its interaction with transcription factors like PGC-1α, and the implications for mitochondrial dysfunction in Parkinson’s disease.
12. Clarify the specificity of HDAC inhibitors in targeting neurodegeneration (Lines 631-638). HDAC inhibitors are broadly discussed without considering isoform-specific effects. Discuss the potential of isoform-selective HDAC inhibitors (e.g., HDAC6 or SIRT1 inhibitors) for targeted therapies, including their differential effects on neuronal survival versus neurotoxicity.
13. Improve discussion on the therapeutic potential of DNMT inhibitors (Lines 647-652). The description of DNMT inhibitors for neurodegenerative diseases is brief and lacks examples of ongoing trials or challenges. Include more information on the limitations of DNMT inhibitors in the context of neurons, referencing ongoing or past clinical trials (e.g., decitabine, azacitidine) and preclinical studies highlighting neurotoxicity concerns.
14. Clarify the clinical relevance of epigenetic biomarkers (Lines 655-664). The potential of epigenetic biomarkers like PSEN1 hypomethylation is mentioned but not sufficiently connected to clinical application. Discuss how epigenetic biomarkers could be incorporated into clinical practice for diagnosing or monitoring disease progression and what challenges remain in translating these findings into routine clinical use.
15. Expand the therapeutic strategies section with examples of combined epigenetic and conventional treatments (Lines 680-682). The review briefly mentions the potential for combining epigenetic and conventional therapies but lacks specific examples. Provide examples of clinical trials or preclinical studies where combined therapies (e.g., epigenetic drugs with cholinesterase inhibitors or L-dopa) have synergistic effects in slowing neurodegeneration.
Author Response
Reviewer 2
- Introduction to such a broad scientific area is very scarce. Epidemiology is larger and mainly oriented to USA data, which does not reflect the international characteristics of IJMS. Enhance the discussion on neuroinflammation and epigenetic regulation (Lines 35-37). Neuroinflammation is a common feature of NDDs but is not linked explicitly to epigenetic regulation. Discuss how epigenetic changes (e.g., in cytokine gene promoters) can drive neuroinflammatory processes in diseases like Alzheimer’s and ALS. Provide examples of critical pathways and potential epigenetic targets for modulating neuroinflammation.
Response: Thanks for the comments. Introductions include epidemiology and pathology of the disease. The author presented a global epidemiology. This review article has chosen to focus primarily on the shared epigenetics and therapeutic strategies among Alzheimer's disease, Parkinson's disease, and amyotrophic lateral sclerosis. We appreciate the complexity of neuroinflammation and its potential contributions to these conditions, but it falls outside the scope of our current discussion.
- Clarification of tau phosphorylation’s role in neurodegeneration (Lines 86-104). The section on tau pathology in Alzheimer’s disease does not sufficiently emphasize the interaction between tau hyperphosphorylation and other cellular processes like autophagy. Discuss the link between tau phosphorylation and autophagy dysfunction, highlighting key kinases involved (e.g., GSK3β) and their potential as therapeutic targets.
Response: Thanks for the comments. We have revised the manuscript and included the suggestions
- Add depth to the discussion on mitochondrial epigenetics in NDDs (Lines 125-132). Mitochondrial dysfunction is mentioned but not linked to specific epigenetic modifications. Include studies that explore the connection between mitochondrial DNA methylation, histone modifications, and neurodegeneration, especially in diseases like Parkinson's and ALS.
Response: Thanks for the comments. We have revised the manuscript and included mitochondrial epigenetics as suggested.
- Expand discussion on environmental factors influencing neuroepigenetics (Lines 162-167). The role of environmental factors in modifying epigenetic landscapes is briefly mentioned without specific examples. Provide examples of environmental stressors (e.g., toxin exposure, diet) and their known effects on epigenetic regulation in the brain. Cite studies showing how these factors can contribute to disease onset or progression.
Response: Thanks for the comments. We have included environmental factors and its role in epigenetic regulation as suggested.
- The review includes miRNAs and their epigenetic role in NDDs (Lines 184-186). It mentions miRNAs but does not provide detailed examples of their role in neurodegenerative diseases. Include specific examples of miRNAs, such as miR-9 or miR-124, which are known to regulate genes involved in synaptic plasticity and neuronal survival in diseases like Alzheimer’s and Parkinson’s.
Response: Thanks for the comments. We have revised the manuscript as suggested.
- Clarification of chromatin remodeling enzymes in epigenetic therapy (Lines 207-212). Chromatin remodeling enzymes are mentioned in the context of neuroepigenetics, but their mechanistic role is not clear. Add details about specific enzymes (e.g., SWI/SNF complexes) and their involvement in regulating transcriptional responses to neurodegenerative stimuli. Discuss their potential as therapeutic targets in NDDs.
Response: Thanks for the comments. We have revised the manuscript as suggested.
- More comprehensive mechanistic details for DNA methylation in neurodegeneration (Lines 245-246). The discussion on the role of DNA methylation in synaptic plasticity and cognition lacks depth. Expand on how specific genes like BDNF, ARC, or EGR1 are involved in learning and memory, integrating more experimental evidence, especially from animal models or human studies.
Response: Thanks for the comments. We have modified the manuscript as suggested.
- Greater focus on the heterogeneity of histone modifications in NDDs (Lines 266-273). The section on histone methylation does not adequately address the diverse functional outcomes of different modifications. Provide more examples of specific methylation marks (e.g., H3K27 versus H3K9) and their differential roles in transcriptional repression versus activation in neurodegenerative disorders.
Response: Thanks for the comments. We have modified the manuscript as suggested
- Address limitations in current single-cell epigenetic profiling technologies (Lines 363-372). The limitations of single-cell bisulfite sequencing are briefly mentioned but could be more thoroughly explored. Discuss the technical challenges in single-cell epigenomics, such as DNA degradation, low coverage, and the difficulty of integrating multi-omics data. Suggest potential solutions or emerging technologies that may overcome these challenges.
Response: Thanks for the comments. We have added the suggested modifications into the manuscript
- Highlight recent advances in three-dimensional chromatin structure analysis (Lines 433-450). The section on chromatin organization techniques, such as Hi-C, lacks recent developments and applications. Add recent studies that utilize advanced 3D chromatin mapping techniques to study neurodegeneration, such as how topologically associated domains (TADs) might reorganize in NDDs, affecting gene expression.
Response: Thanks for the comments. We have revised the manuscript as suggested
- Clarification on the role of α-synuclein in histone modifications in Parkinson's Disease (Lines 511-521). The impact of α-synuclein on histone acetylation is noted but underdeveloped. Elaborate on the specific molecular pathways through which α-synuclein interferes with histone acetylation, particularly its interaction with transcription factors like PGC-1α, and the implications for mitochondrial dysfunction in Parkinson’s disease.
Response: Thanks for the comments. We have revised the manuscript as suggested
- Clarify the specificity of HDAC inhibitors in targeting neurodegeneration (Lines 631-638). HDAC inhibitors are broadly discussed without considering isoform-specific effects. Discuss the potential of isoform-selective HDAC inhibitors (e.g., HDAC6 or SIRT1 inhibitors) for targeted therapies, including their differential effects on neuronal survival versus neurotoxicity.
Response: Thanks for the comments. We have added the suggested modifications into the manuscript
- Improve discussion on the therapeutic potential of DNMT inhibitors (Lines 647-652). The description of DNMT inhibitors for neurodegenerative diseases is brief and lacks examples of ongoing trials or challenges. Include more information on the limitations of DNMT inhibitors in the context of neurons, referencing ongoing or past clinical trials (e.g., decitabine, azacitidine) and preclinical studies highlighting neurotoxicity concerns.
Response: Thanks for the comments. We have added the suggested modifications into the manuscript
- Clarify the clinical relevance of epigenetic biomarkers (Lines 655-664). The potential of epigenetic biomarkers like PSEN1 hypomethylation is mentioned but not sufficiently connected to clinical application. Discuss how epigenetic biomarkers could be incorporated into clinical practice for diagnosing or monitoring disease progression and what challenges remain in translating these findings into routine clinical use.
Response: Thanks for the comments. We have revised the manuscript as suggested
- Expand the therapeutic strategies section with examples of combined epigenetic and conventional treatments (Lines 680-682). The review briefly mentions the potential for combining epigenetic and conventional therapies but lacks specific examples. Provide examples of clinical trials or preclinical studies where combined therapies (e.g., epigenetic drugs with cholinesterase inhibitors or L-dopa) have synergistic effects in slowing neurodegeneration
Response: Thanks for the comments. We have added the suggested modifications into the manuscript
Round 2
Reviewer 1 Report
Comments and Suggestions for Authors
the authors did not make any changes regarding multiple sclerosis and there is no reason why the authors simply ignored the comment.
Author Response
We highly appreciate the reviewer’s suggestion and apologize for the oversight. The recommended modifications have now been implemented as suggested, enhancing the clarity and alignment of the manuscript with your feedback. Thank you for helping us improve our work.

Reviewer 2 Report
Comments and Suggestions for Authors
The authors have consistently improved their manuscripts, which I consider can be accepted now.
Author Response
We highly appreciate reviewer's suggestion in improving our manuscript.
Round 3
Reviewer 1 Report
Comments and Suggestions for Authors
MS is not only demyelinating disease but neurodegenerative as well, and the authors ignore that fact.
However, I am not willing to comment anymore.